# Chromatin interactions and expression quantitative trait loci reveal genetic drivers of multimorbidities

Tayaza Fadason [1,2], William Schierding[1,2], Thomas Lumley[3] & Justin M. O'Sullivan [1,2]

Clinical studies of non-communicable diseases identify multimorbidities that suggest a common set of predisposing factors. Despite the fact that humans have ~24,000 genes, we do not understand the genetic pathways that contribute to the development of multimorbid non-communicable disease. Here we create a multimorbidity atlas of traits based on pleiotropy of spatially regulated genes. Using chromatin interaction and expression Quantitative Trait Loci (eQTL) data, we analyse 20,782 variants ($p < 5 \times 10^{-6}$) associated with 1351 phenotypes to identify 16,248 putative spatial eQTL-eGene pairs that are involved in 76,013 short- and long-range regulatory interactions (FDR < 0.05) in different human tissues. Convex biclustering of spatial eGenes that are shared among phenotypes identifies complex interrelationships between nominally different phenotype-associated SNPs. Our approach enables the simultaneous elucidation of variant interactions with target genes that are drivers of multimorbidity, and those that contribute to unique phenotype associated characteristics.

[1] The Liggins Institute, The University of Auckland, Auckland 1023, New Zealand. [2] Maurice Wilkins Centre for Molecular Biodiscovery, Auckland 1010, New Zealand. [3] The Department of Biostatistics, The University of Auckland, Auckland 1010, New Zealand. Correspondence and requests for materials should be addressed to J.M.O. (email: justin.osullivan@auckland.ac.nz)

The rising incidences of complex diseases and their accompanying cost burdens[1,2] are driving a shift in disease research and management from the single-disease to a broader paradigm that accommodates a patient's overall health[2,3]. At the same time, there is great expectation that personalised medicine will aid in delivering medical care that is more suitable to the individual. Completing this paradigm shift requires many advances including a greater understanding of the genetic aetiology of the observed multimorbidities, which remain largely unknown[4]. The importance of understanding the genetic aetiology of multimorbidity is further compounded by the fact that almost 90% of single-nucleotide polymorphisms (SNPs)—a major source of shared heritability of polygenic disorders—do not fall within coding regions of the genome[5].

Cross-phenotype genetic studies have been conducted on a small number of complex traits (the largest one studied 42 traits[6]) with known associations using methods that included systematic reviews[7], LD score regression[8,9], polygenic risk scores[10], Probabilistic Identification of Causal SNPs[11] and Bayesian colocalisation tests[6] on Genome Wide Association Study (GWAS) summary or molecular data. These studies[6–8,10,11] typically use SNPs, or the genes that are nearest to—or in LD with—the SNPs as the putative genetic drivers of cross-phenotype associations. These approaches are limiting as evidence increasingly shows that gene regulatory elements (e.g. enhancers) can impact distant genes more strongly than the genes harbouring, or closest to, them as a result of physical interactions with distal chromatin regions within the 3D organisation of the genome[12–18].

In this study, we undertake a discovery-based approach to identify spatial eQTL-eGene pairs (i.e. phenotype-associated SNP-gene pairs that are supported by both interaction and eQTL data) for all human traits within the GWAS Catalog. Our approach identifies SNP-gene regulatory relationships, ~75% of which are missed by proximity in the GWAS Catalog associations. The integration of spatial data allows for the identification of trans-eQTL associations. Using a convex biclustering algorithm, we identify clusters of multimorbidities according to shared target eGenes of traits. The loci at the centre of the resulting phenotype clusters are subject to complex tissue and disease specific regulatory effects. The largest cluster, 40 phenotypes that are related to fat and lipid metabolism, inflammatory disorders, and cancers, is centred on the *FADS1-FADS3* locus (chromosome 11). Lastly, we show that eQTLs marked by common variants also have a regulatory role in rare Mendelian disorders. Our results demonstrate the utility of this approach in understanding the common genetic aetiology of multimorbid traits.

## Results

**GWAS SNPs mark spatial regulatory regions**. Disease-associated SNPs often mark gene enhancers, silencers, and insulators[19]. We set out to identify the genes whose transcript levels are associated with regulatory regions marked by disease- and phenotype-associated SNPs (daSNPs). We downloaded 20,782 daSNPs ($p < 5 \times 10^{-6}$) from the GWAS Catalog (www.ebi.ac.uk/gwas/). The CoDeS3D pipeline[14] was used to interrogate Hi-C chromatin interaction libraries[20] (see Methods) to identify genes that are captured as physically interacting with the GWAS SNP-labelled regions (Fig. 1a) within nuclei from one or more of seven cell lines (GM12878, HMEC, HUVEC, IMR90, K562, KBM7 and NHEK)[20]. The resulting 1,183,037 spatial SNP-gene pairs were used to query the GTEx database (www.gtexportal.org, multi-tissue eQTLs analysis v4) to identify spatial eQTL-eGene pairs, out of which only 16,248 SNP-gene pairs had significant (FDR ≤0.05) eQTL associations. Spatial SNP-gene pairs with evidence of interaction in >1 cell lines or >1 replicates in a single

cell line are significantly more enriched (two-proportions Z-test, $p$-value $<2.2 \times 10^{-16}$) for eQTL associations than pairs with only one interaction in one replicate of a cell line (Fig. 1b).

A total of 7776 (~38.4%) of the GWAS SNPs analysed were associated with a change in the expression (i.e. eQTLs) of 7917 eGenes at an FDR ≤ 0.05 (Benjamini Hotchberg[14]), for a total of 16,248 distinct spatial eQTL-eGene pairs (76,013 interactions in different tissues). daSNPs with significant eQTLs are distributed (range = 86–764) across chromosomes 1-22. The distributions of daSNPs and spatial eQTLs correlate (Pearson's $r$, 0.86 and 0.74, respectively) with the sizes of the chromosomes, with chromosomes 1 and 21 having the most and least spatial eQTLs, respectively (a in Supplementary Fig. 1). Similarly, the number of spatial eGenes on chromosomes correlated (Pearson's $r = 0.86$) with the number of genes per chromosome (b in Supplementary Fig. 1). Notably, none of the 182 daSNPs on chromosome X was identified as having significant spatial eQTL effects with any gene (b in Supplementary Fig. 1). However, expression of 41 genes on the X chromosome is associated with spatial eQTLs from other chromosomes (b and d in Supplementary Fig. 1). The under-representation of GWAS SNPs and spatial eQTLs on the X chromosome can be explained by the exclusion of X-linked genetic variants from GTEx and two-thirds of GWAS[21,22]. As none of the databases referenced here have Y or mitochondrial data, there are also no SNPs nor spatial eGenes identified on those chromosomes.

Only 24.3% of the spatial eQTL-eGene pairs matched the SNP-gene mapping in the GWAS Catalog (Fig. 1c). 13,240 (75.7%) spatial eQTL-eGene pairs are missed in the GWAS mapping of genes to SNPs. Notably, the inclusion of spatial information was associated with a similar increase in the detection of tissue-specific eQTL-eGene regulation (Fig. 1c). This finding is consistent with previous observations[14,22] that highlighted the discordant results between the nearest-gene and eQTL-based assignment of GWAS SNPs to target genes. Of the 7917 affected eGenes, 70.0% (5545) were associated with only *cis*-spatial interactions (i.e. both partners are from the same chromosome and separated by <1,000,000 bp, as defined elsewhere[22,23]), 19.3% (1528) by only *trans*-interactions (i.e. 663 are affected by an eQTL SNP on the same chromosome but separated by ≥1,000,000 bp, and 865 are affected by an eQTL SNP from different chromosomes[22,23]), and 10.7% (844) by both *cis*- and *trans*-interactions (Fig. 1c, a in Supplementary Fig. 2, Supplementary Data 1). Approximately 49% of spatial eQTL SNPs affect more than one gene (b in Supplementary Fig. 2). The spatial eQTL-eGene Hi-C fragment loop distances range from 0 bp to 248 Mb (c in Supplementary Fig. 2). The most significant spatial eQTL-eGene interactions involved associations between: (1) the long intergenic non-protein coding RNA *CRHR1-IT1* (Chr. 17) and rs12373124, rs12185268, rs1981997, rs2942168, rs17649553, rs17689882 and rs8072451 in subcutaneous adipose tissue ($p_{eQTL}$ range, $1.39 \times 10^{-95}$–$4.33 \times 10^{-94}$); and (2) *PEX6* (Chr. 6) and rs9296404 in skeletal muscle ($p_{eQTL}$ $1.59 \times 10^{-93}$; c and d in Supplementary Fig. 1).

To estimate how much functional information is gained by the integration of spatial data, we took all 339 daSNPs on chromosome 22 and queried the GTEx v7 analysis for significant eQTL associations. The GTEx-only method yielded 4408 eQTL associations, all of which were within 1 Mb genomic distance (Fig. 1d, e, Supplementary Data 2). We then analysed the same set of 339 daSNPs using the CoDeS3D pipeline, which integrates GTEx v7 analysis. The CoDeS3D approach identified 4543 spatial eQTL associations of which 3542 (~78%) were also found in the GTEx-only associations (Fig. 1d, e, Supplementary Data 2). 866 (19.6%) of the GTEx-only associations, with eQTL normalised effect sizes (NES, i.e. the slope of linear regression) of −1.11 to 1.15, were lost in the CoDeS3D analysis due to lack of evidence for a spatial

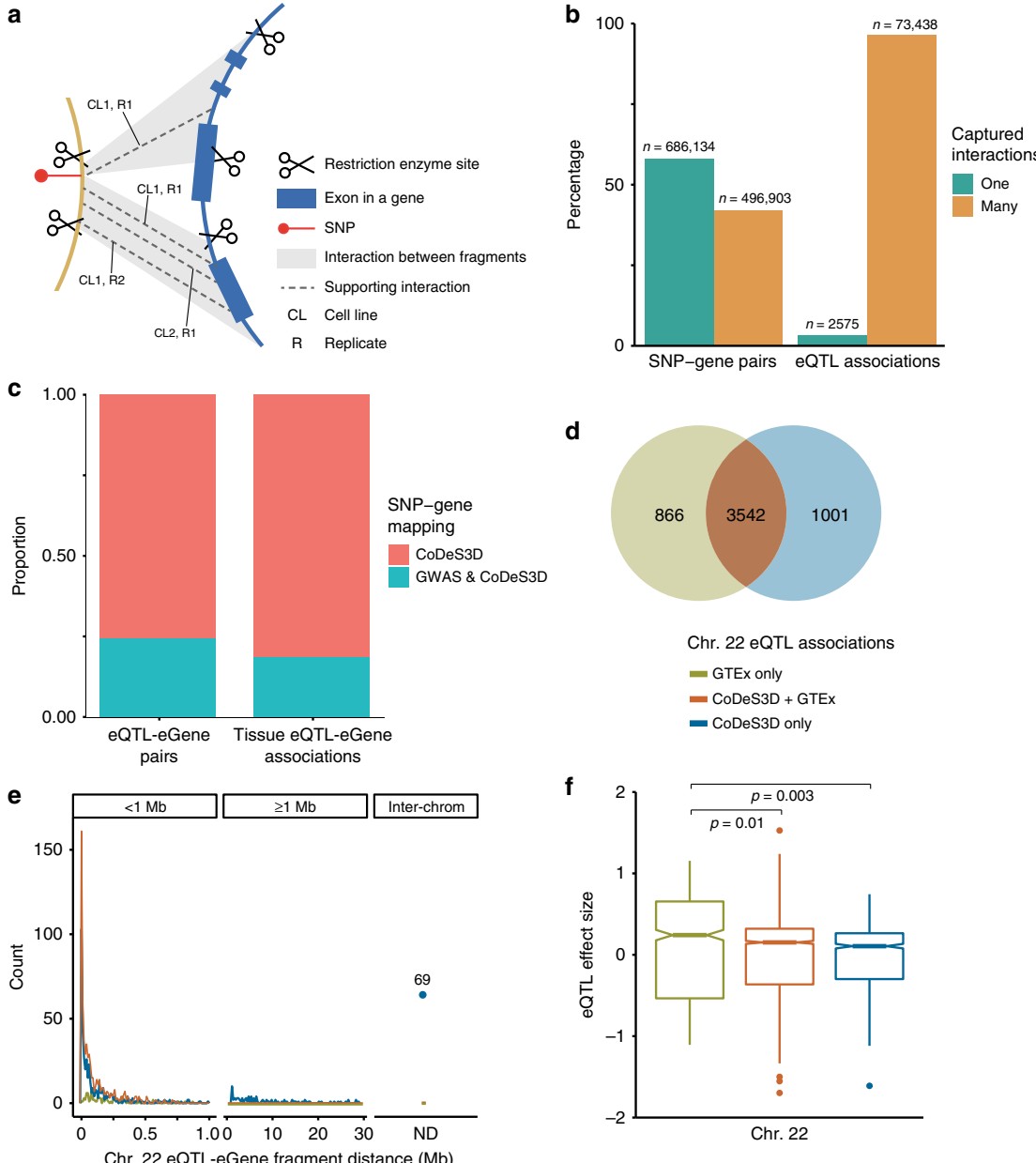

**Fig. 1** Integration of interaction data identifies distant target genes of GWAS SNPs. A spatial SNP-gene pair is defined as interaction(s), in more than one cell line, between the fragment harbouring the SNP and any fragment(s) in the region spanning the gene. In the cartoon example in **a** there are two unique fragment interactions between the SNP and gene fragments. One interaction is captured in only one replicate, R1, of cell line, CL1. The second interaction is captured in two replicates, R1 and R2, of cell line CL1, and one replicate, R1, of cell line CL2. Thus, the SNP-gene pair has a total of two fragment interactions, and four supporting interactions. **b** Genome-wide SNP-gene pairs with ≥1 supporting interactions are enriched (two-proportions Z-test p-value < 2.2 × 10$^{-16}$) for eQTL association. **c** Integration of spatial data enabled the identification of genome-wide eQTL-eGene pairs that were missed by GWA studies, which typically used linear proximity to identify the target gene. It is notable that the identification of eQTL-eGene pairs extended to tissue-specific effects. **d** The majority of eQTL associations on chromosome 22 are identifiable by incorporating spatial information (i.e. Hi-C interaction data). Calculation of eQTLs using GTEx alone identified 866 GTEx-only associations. By contrast, CoDeS3D identified 1001 associations that had a spatial component. **e** Integration of spatial data allows for detection of distal eQTL associations on chromosome 22. ND not determinable. **f** The mean eQTL normalised effect size (NES) of the 1001 spatial eQTL informed associations, on Chr. 22, is significantly different from the mean eQTL effect size of the 866 associations from only GTEx. Centre line, bounds of box, and whiskers of boxplots represent the median, 2nd and 3rd quartile, and minimum and maximum values, respectively

connection between the genomic fragments containing the SNP and gene. Conversely, the integration of spatial data enabled the detection of 1001 spatial eQTL associations, with eQTL NES ranging between −1.61 and 0.74, that were not detected by the GTEx-only method (Fig. 1d, f). These results are consistent with a significant number of eQTL effects involving 3D looping interactions, up to and beyond 1 Mb. The absence of direct physical

contacts for the 866 eQTLs is consistent with alternative mechanisms, including the diffusion of regulatory factors released from the eQTL locus[13], contributing to the regulatory network.

**Multimorbid phenotypes cluster around shared eGenes**. We reasoned that the common pathogenesis seen in polygenic

diseases and phenotypes occurs because they share common biochemical pathways and genes. To identify associations among phenotypes, we populated phenotype matrices according to their common spatial eQTL SNPs or eGenes (a and b in Supplementary Fig. 3, Supplementary Data 3). Q–Q plots of the ratios of shared spatial eQTL SNPs and eGenes identified greater coverage of intermediate values between 0 and 1 that was consistent with increased information within the phenotype matrix that was populated using the eGenes (a and b in Supplementary Fig. 3). To ensure that the spatial eGene association pattern was non-random, we generated 1000 null datasets by pooling together all phenotype-associated spatial eGenes and randomly reassigning them to phenotypes, such that each control phenotype had the same number of eGenes as its corresponding test phenotype. Phenotype matrices populated by the mean null phenotype eGenes ratios had different association and distribution patterns from those generated for the test phenotypes (c in Supplementary Fig. 3, Supplementary Data 3).

Of the 1351 phenotypes analysed, 618 are significantly associated with ≥4 spatial eGenes. Convex biclustering[24] revealed the intricate interrelationships among the 618 phenotypes based on spatial eGene pleiotropy (Supplementary Fig. 4). Phenotype clusters included: (a) closely related measures of a phenotype (e.g. hypertension, blood pressure and pulse pressure); (b) phenotypes which are observed as co- or multimorbid (e.g. white matter hypersensitivities, stroke and dementia[25]; or ovarian cancer, interstitial lung disease, Alzhiemer's disease and other cognitive disorders[26–28]); and (c) phenotypes that have controversial reports of inter-phenotype association e.g. autism spectrum disorder and iron biomarker levels[29,30]. Notably, the observed pattern of multimorbidity derived from shared spatial eGenes is different from the interrelationships between phenotypes with ≥4 spatial eQTL SNPs (Supplementary Fig. 5).

Multiple variants from one genomic region have previously been associated with cross-phenotypes e.g. the *IFI30* locus in autoimmune diseases[11], and the *CDKN2B-AS1* locus in coronary artery disease, glioma and intracranial aneurysm[31]. However, these studies did not resolve the target genes of the variants. It is noteworthy that spatial eGenes that are common to most phenotypes in the clusters we detected lie adjacent to each other in a contiguous genomic region (typically 100–400 kb in length) and are in cis-association with the eQTL SNPs. This is exemplified by a subset of immune-related disorders that cluster about the *PGAP3–GSDMA* locus (Chr. 17: 37,827,375–38,134,431; hg19); skin pigmentation and skin cancer, which are clustered about the *SPATA33–URAHP* region (Chr. 16:89,724,152–90,114,191; hg19); and a mood disorder cluster that is built about the *NT5DC2-TMEM110* locus (Chr. 3:52558385–52931597; hg19; Supplementary Fig. 6). This study, to the best of our knowledge, is the first to observe contiguous target genes of spatial eQTL SNPs in complex cross-phenotypes.

The largest observed multimorbid cluster (Supplementary Fig. 4, cluster#6) is an outgroup of phenotypes located in the bottom left hand corner of the matrix that highlights inter-relationships among polyunsaturated fatty acids (PUFAs), Crohn's disease, inflammatory bowel disease, colorectal cancer, laryngeal squamous carcinoma, insulin sensitivity, comprehensive strength index, and cholesterol levels (Fig. 2a). The cluster is built about a 283 kb region on chromosome 11 that contains the *DAGLA, MYRF, TMEM258, RP11-467L20, FADS1, FADS2, FADS3,* and *BEST1* genes (Fig. 2b; Supplementary Data 4). Consistent with this observation, genetic variations in the *FADS1-FADS3* region have previously been associated with alterations in the synthesis of PUFAs[32], inflammatory bowel diseases[33], cholesterol levels and BMI[34], coronary artery disease and type 2 diabetes[35], and colorectal cancer[36]. Our discovery-based

approach also confirms earlier observations of pulmonary multimorbidity and genetically controlled regulatory variation in the *CHRNA* region[37] (a in Supplementary Fig. 6 and Supplementary Fig. 9).

**Common genes are affected by different disease eQTLs**. We mapped the spatial eQTL-eGene interactions within the *FADS1-3* locus (Fig. 3a) in order to investigate the effect of genetic variation on the regulatory network for the multimorbid phenotypes associated with the cluster. The transcription levels of the *FADS1, FADS2, TMEM258 and DAGLA* genes, that are central to this cluster, are associated with eQTLs that are located within these genes and across the region (Fig. 3a). Nine of the putative regulatory regions are located within introns of genes (i.e. *MYRF, TMEM258, FEN1, FADS1, FADS2 and FADS3*) in this locus. Putative regulatory effects linking eQTLs in *FADS1-3* with *DAGLA*, or eQTLs in *MYRF* with *FADS1-3* cross a topologically associating domain (TAD) boundary located in the vicinity of *FEN1*, whose transcription is not associated with any of the eQTLs (Supplementary Fig. 7). Spatial eQTLs associated with some phenotypes (e.g. LDL cholesterol, muscle measurement and comprehensive strength index) are few and localised while others (e.g. cis-trans-18:2 fatty acid, phospholipid) are dispersed across the locus. However, despite this, almost all of the phenotype-associated SNPs in this cluster are correlated with a change in the transcript level of more than one gene (Fig. 3b). We also observed this one-to-many SNP-eGene eQTL association in the pulmonary cluster, including inter-TAD connections from the region marked by rs8042374 (c in Supplementary Fig. 9). Collectively, these results are consistent with previous reports of the formation of complex networks of multiple long-range interactions by regulatory elements and gene promoters[38].

**eQTLs have gene- and tissue-specific effect patterns**. SNPs that are in high linkage disequilibrium (LD) might be predicted to have inseparable regulatory effects on target genes. However, given the composite nature of regulatory elements and networks, it is likely that even linked SNPs affect different regulatory elements. Therefore, we obtained the effect sizes of spatial eQTLs within the PUFA eGene cluster from different tissues (GTEx v7, 01/12/2017) to identify associations between eQTLs that are in strong LD. We characterised two distinct patterns, i.e. groups A and B, of eQTL associations with the target genes within the *FADS* region (Supplementary Data 5). The tissue eQTL effect patterns of the linked SNPs seem to be consistent with their differences in allele frequency, as informed by the $R^2$ and $D'$ scores (Fig. 4, a in Supplementary Fig. 8, Supplementary Data 5). However, some exceptions exist e.g. in the CEU population rs174574 (minor allele frequency, MAF, 0.63) in Group B and rs1535 (MAF = 0.36) in Group A are in complete LD ($R^2 = D' = 1$) but have opposite effects on their common target genes. The minimum and maximum effect sizes of rs1535 are $-0.76$ (*FADS1*, $p_{eQTL} = 5.3 \times 10^{-21}$, cerebellum) and 0.8 (*FADS2*, $p_{eQTL} = 2.3 \times 10^{-10}$, spleen, while the maximum and minimum effect sizes of rs174574 are $-0.72$ (*FADS2*, $p_{eQTL} = 1.5 \times 10^{-8}$, spleen) and 0.73 (*FADS1*, $p_{eQTL} = 1.5 \times 10^{-18}$, cerebellum). By contrast, rs1000778 (MAF = 0.73) and rs422249 (MAF = 0.65) have a similar eQTL effect pattern (Group B) on target genes, *FADS1* and *FADS2* despite lower levels of linkage (CEU $R^2 = 0.365$, $D' = 0.753$). We observed similar patterns linking LD and regulatory effects within the *CHRNA* locus (a and b in Supplementary Fig. 9). The minimum CEU $D'$ and $R^2$ scores for the eQTL SNPs in the *CHRNA* locus is 0.715 and 0.097, respectively. Again, the tissue eQTL effect pattern is similar to the $R^2$ scores, which are allele frequency-dependent LD measures.

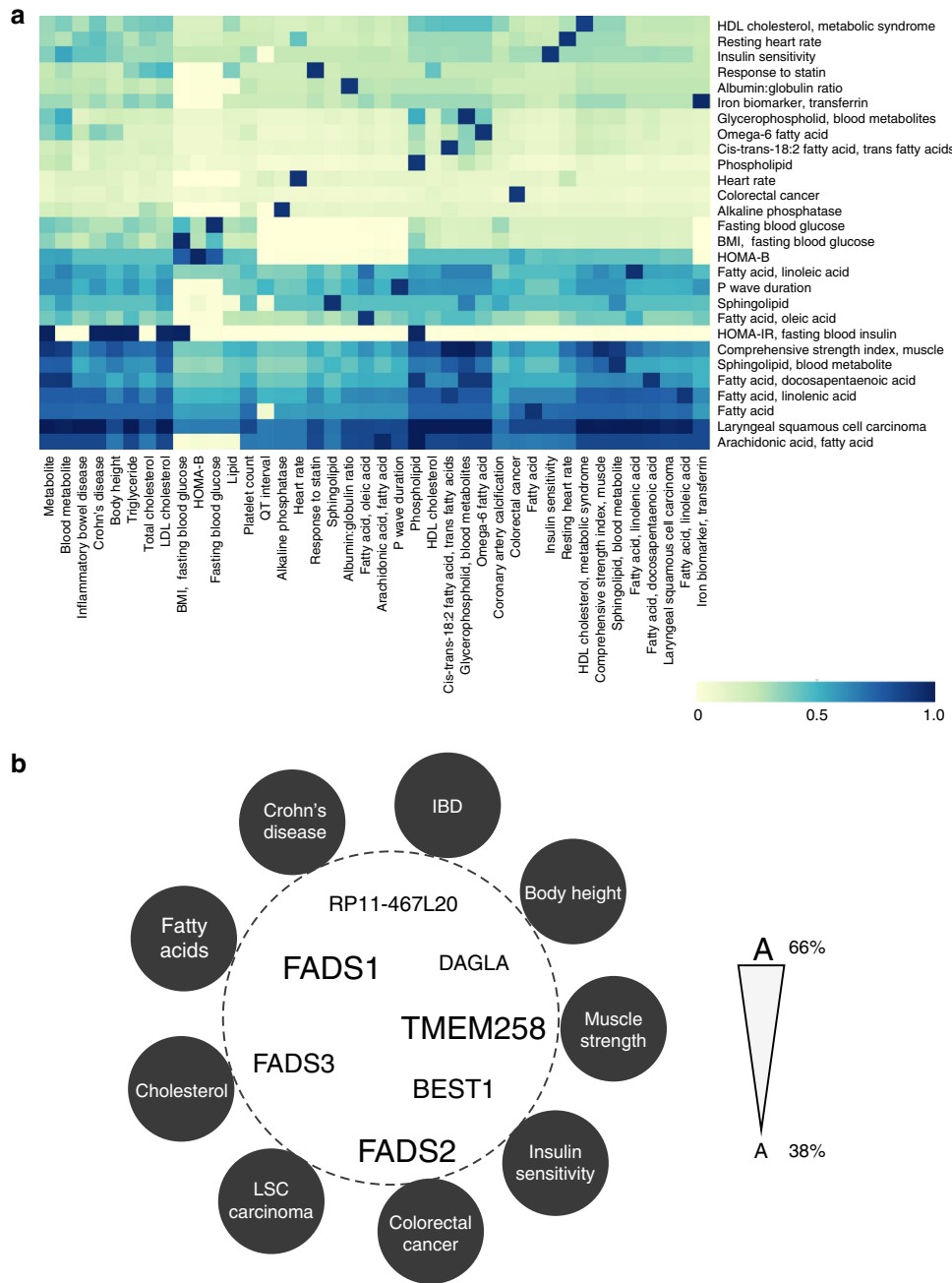

**Fig. 2** The largest cluster comprises 40 phenotypes related to fat metabolism. **a** SNPs associated with inflammatory bowel disease, cholesterol levels, insulin sensitivity, muscle strength index and laryngeal squamous cell carcinoma, amongst others, are associated with transcription effects at a pool of shared spatial eGenes. Deep blue squares indicate higher proportions of shared eGenes, with 1 being the highest and indicating that two phenotypes have the same set of eGenes. **b** The *FADS* locus on chromosome 11 is responsible for the identified interrelationships (i.e. the multimorbidity) between the phenotypes in this cluster. We hypothesise that the interrelationships between the additional 2431 spatial eGenes that are associated with these multimorbid phenotypes contribute to the unique characteristics and sub-clustering of the phenotypes (Supplementary Data 4). In this word cloud, the size of a gene name represent the percentage of phenotypes whose associated variants spatially affect the expression of that gene in the cluster e.g. the expression of *FADS1*, *FADS2* and *TMEM258* are associated with eQTLs in 26 of the 40 phenotypes in the cluster (Supplementary Data 4)

The spatial eQTLs in the fat metabolism cluster that are located within the *FADS1-3* locus are all associated with an inverse relationship between *FADS1* and *FADS2* transcription levels (Fig. 4, Supplementary Fig. 8). At the time of writing our results, these inverse eQTL effects on *FADS1* and *FADS2* were also reported in another study[39]. Our results reveal that *FADS1* and *FADS2* are not the only genes affected in this region (b in Supplementary Fig. 8). Wherever there is an eQTL effect on *MYRF*, *FEN1*, and *FADS3*; the direction of the effect size is the

same as observed for *FADS1*. On the other hand, eQTL effects on *TMEM258* mirror those that occur at *FADS2*. Notably, all but five eQTLs (i.e. rs174574, rs422249, rs174448, rs174449, and rs100078) are associated with a negative effect on *FADS1* and a positive effect on *FADS2* transcript levels (b in Supplementary Fig. 8). It is possible that the opposite regulatory effect observed for these five eQTLs represents allele flipping (Supplementary Data 5). Our results indicate that a composite regulatory hub forms from dispersed locations to regulate the convergent,

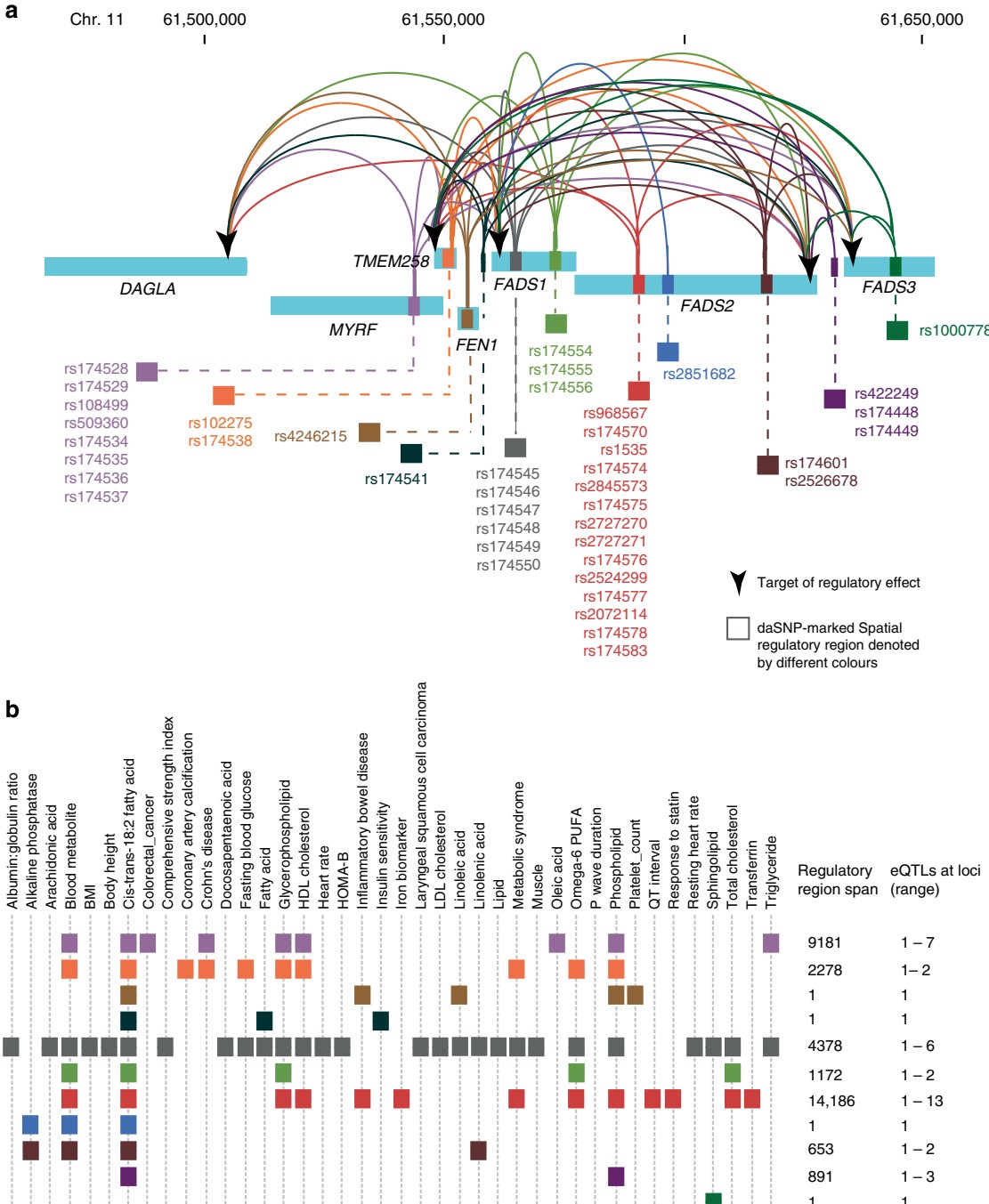

**Fig. 3** Spatial eQTL-eGene interactions central to the fat metabolism cluster. **a** Transcript levels for genes located within Chr. 11: chr11:61447905–61659017 are associated with eQTLs located in clusters across the 283 kb locus. For simplicity, we grouped eQTLs according to separation in the linear sequence such that they are located in different genes, or are separated by ≤5 kb. Genomic locations are from human genome Hg19 and the eQTL-eGene interaction analysis used GTEx v4 (18/10/2016). **b** Phenotype-associated eQTLs are localised or dispersed across the *FADS1* locus. eQTLs associated with cis-trans-18:2 fatty acid levels are the most dispersed. eQTLs rs174545-50 are associated with the most phenotypes. The regulatory region span (in bp) includes the spatial eQTL SNPs in the group. eQTL SNPs are coloured according to the putative regulatory regions in **a**

duplicated (61% amino acid identity and have 75% similarity[32]) *FADS1* and *FADS2* genes.

Consistent with our understanding of tissue and cell type specificity of gene regulation, the tissue eQTL associations showed cell line-dependent enrichment patterns (Fig. 5a). The strongest and weakest eQTL associations in subcutaneous adipose were observed in the GM12878 and KBM7 cell lines respectively, while in omental visceral adipose, they were observed in the HMEC and NHEK cell lines. There was no correlation between the total strength (i.e. *p*-values) of eQTL associations and the total number of spatial interactions (a and b in Supplementary Fig. 10). Similarly, the observed frequency of Hi-C fragment contact counts of the eQTL-eGene pairs in the cell lines showed less tissue specificity (Fig. 5b; c in Supplementary Fig. 10). There was a strong positive correlation ($r = 0.87$) between the percentage of spatial eQTL-eGene interactions and the number of RNAseq GTEx samples in tissues (d in Supplementary Fig. 10). Collectively, these results suggest that the frequency of spatial

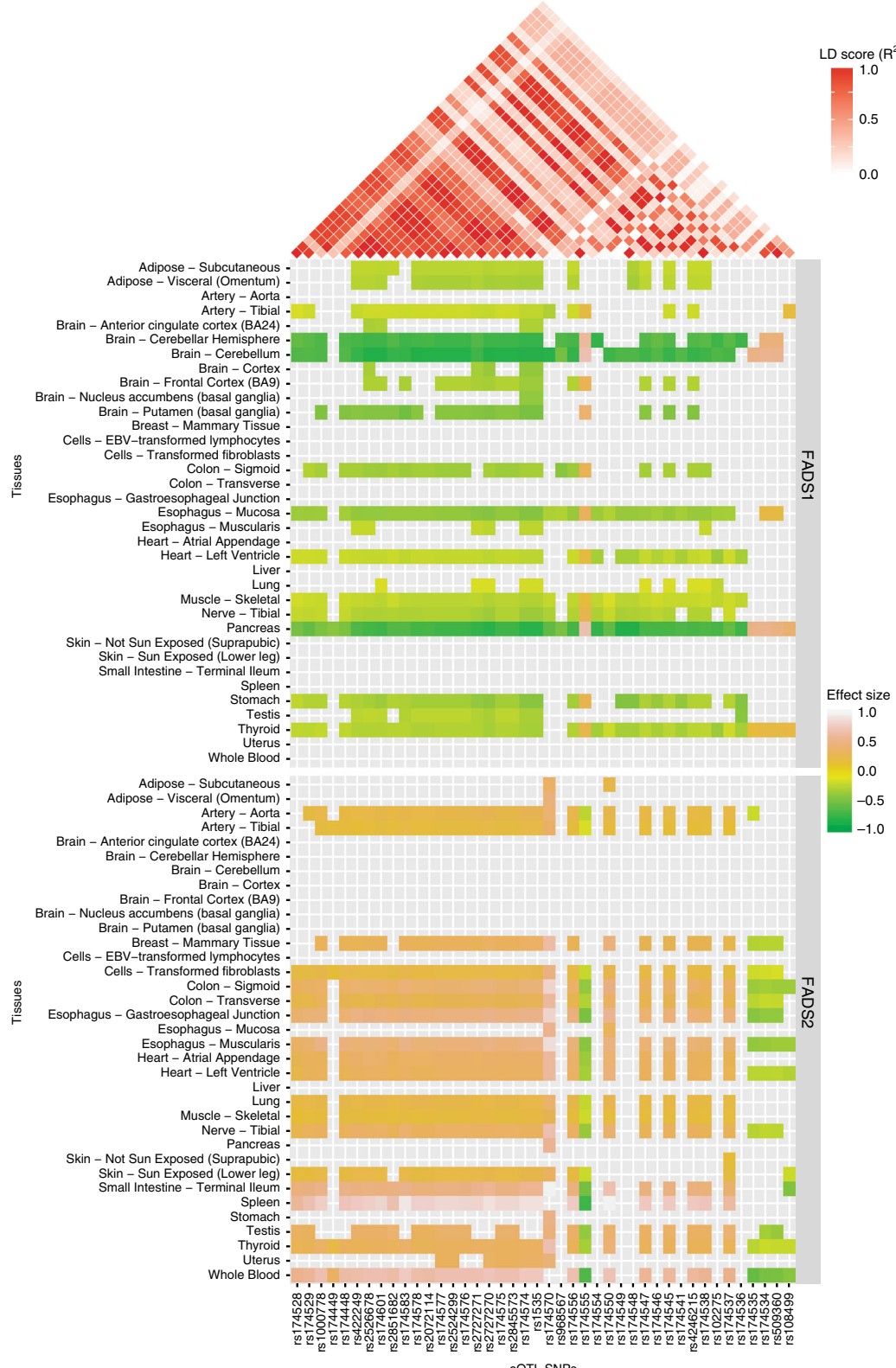

**Fig. 4** eQTLs have different effect patterns within the fat metabolism cluster. Spatial eQTLs have tissue-specific and LD-dependent effects on genes in the *FADS* cluster. There is also an inverse association between the eQTL effects on *FADS1* and *FADS2*. rs174574, rs422249, rs174448, rs174449, and rs100078 are associated with an increase in *FADS1* and decrease in *FADS2* transcript levels. All other eQTLs are associated with a decrease in *FADS1* and increase in *FADS2* transcript levels. Effect sizes of spatial eQTL on eGenes were obtained from GTEx v7

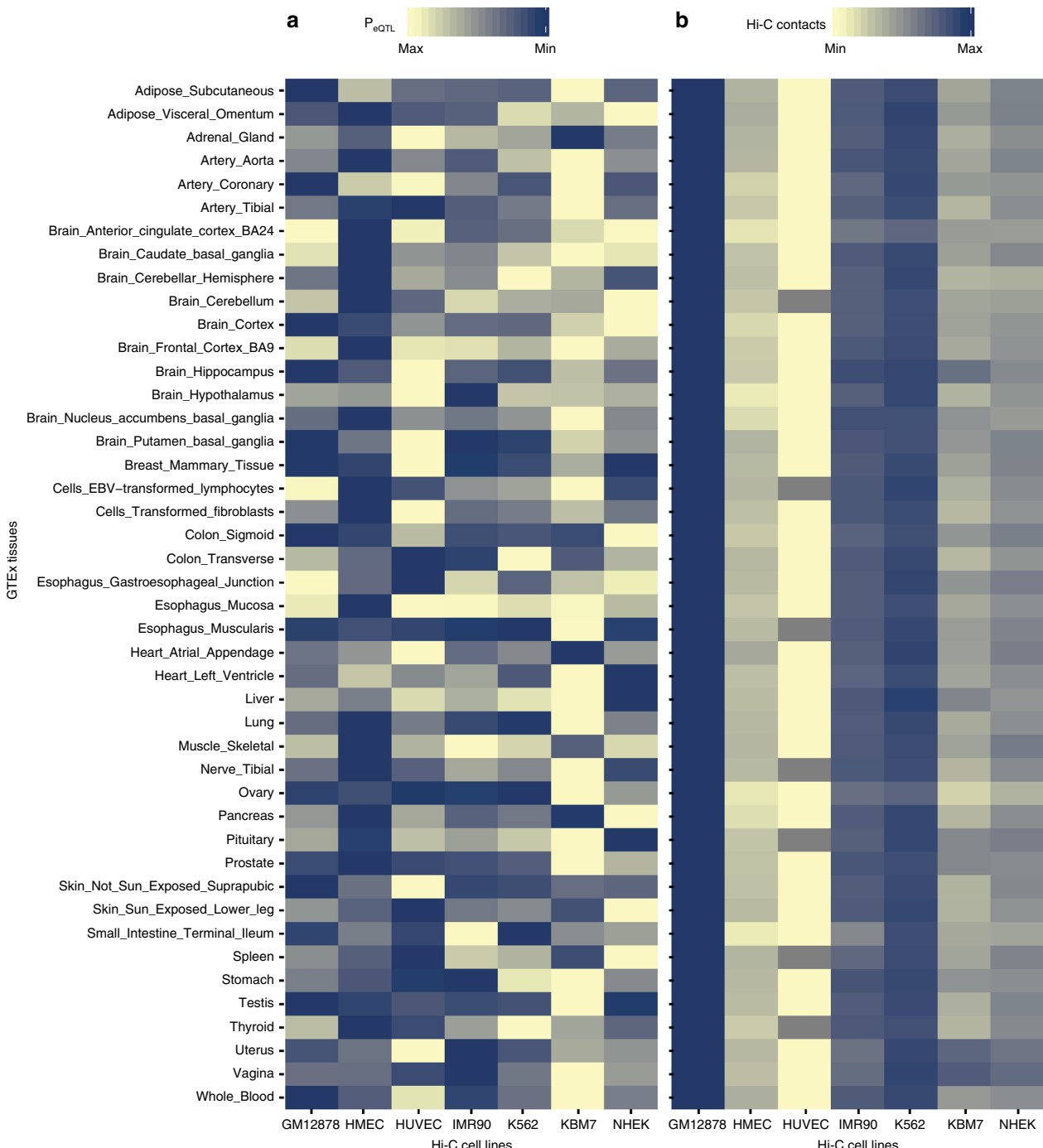

**Fig. 5** Tissue and cell line-specific effects of eQTLs. **a** eQTL effects are strong in Hi-C cell lines that represent the tissues that they are derived from. The heatmap shows the range of mean eQTL *p*-values in tissues, with the yellow and blue colours representing the Hi-C cell lines with the minimum and maximum mean eQTL p-values respectively. **b** The number of Hi-C contact counts between the regions containing the eQTL-eGene pairs show less tissue specificity

eQTL-eGene contact in cell lines are relatively uniform across tissues but the eQTL effects in the cell lines are tissue specific. This is consistent with previous results that show that transcription is not necessary for the formation and maintenance of a spatial connection[40,41].

**GWAS eQTLs spatially affect Mendelian genes.** Rare monogenic or Mendelian diseases are typically considered to be associated with highly penetrant loss of function mutations. However, genes linked to Mendelian diseases have also been implicated in polygenic disorders[42,43]. Therefore, we determined if the spatial eGenes we identified as being associated with complex diseases, were also implicated in Mendelian diseases. Using the OMIM database, a catalogue of associations between human genes and mendelian traits, we identified that 62% (5069) of the spatial eGenes are catalogued in the OMIM database (Fig. 6a, Supplementary Data 6). This is consistent with the possibility that there is a distal regulatory component in rare Mendelian disorders.

There are four gene-phenotype mapping categories in the OMIM database: (1) the unknown defect category, in which a

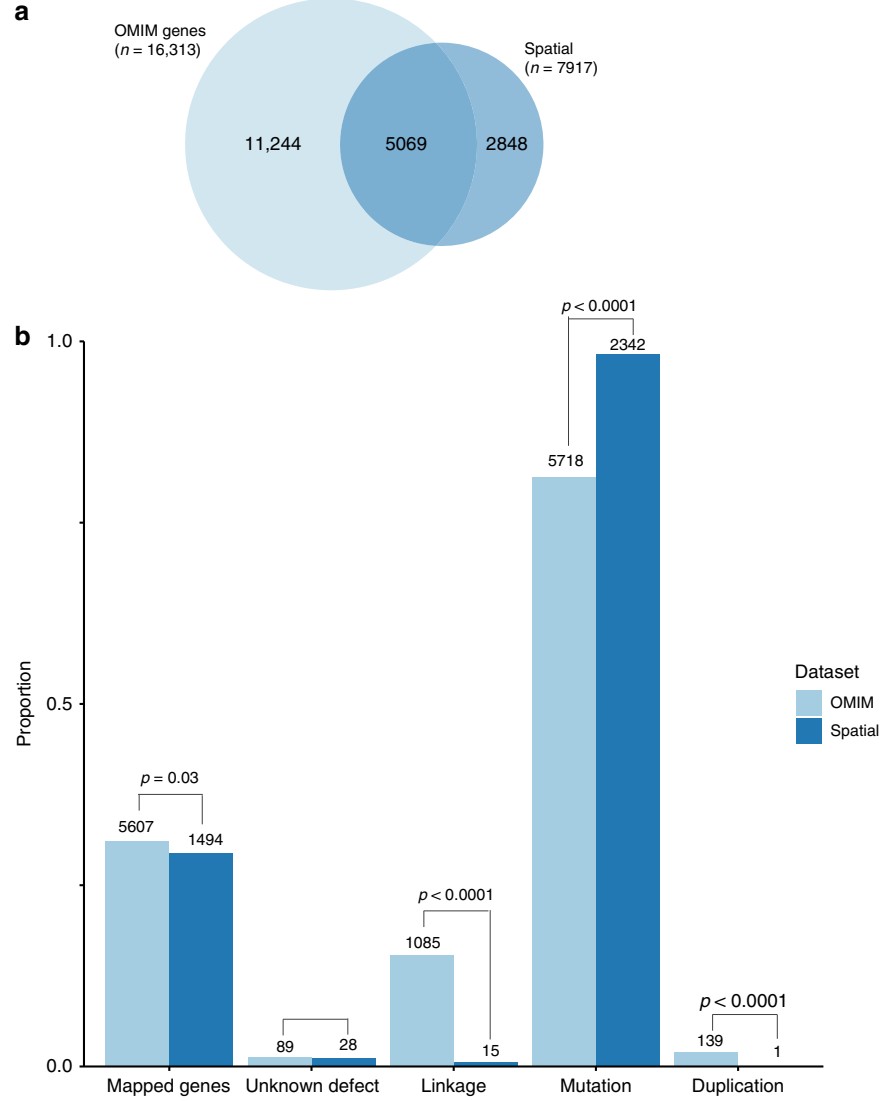

**Fig. 6** OMIM analysis of spatial eQTL-eGene interactions. **a** A total of 62% of the spatial eGenes we identified are associated with human disease in the OMIM database (retrieved, 11/08/2017, Supplementary Data 6). **b** A total of 98.2% of gene-phenotype associations of the identified spatial eGenes are based on known mutations in the genes. OMIM mapping methods are: (1) gene has unknown underlying defect but is associated with the disorder; (2) disorder is mapped to gene based on linkage but mutation in gene has not been found; (3) a mutation in the gene has been identified as the basis of the mapped disorder; and (4) disorder is caused by deletion or duplication of contiguous genes. Mapped gene proportions were calculated as the number of genes mapped to at least one phenotype in the OMIM data (annotated in graph) divided by the total number of genes in the OMIM data (16,313 for OMIM genes) or the number of spatial eGenes annotated in the OMIM data (5069 for eGenes)

disorder is associated with a gene with no known functional role; (2) the linkage category, which maps a gene to a disorder based on linkage despite no mutation having been found in the gene yet; (3) the mutation category, in which a mutation in the gene has been identified as the basis of the mapped disorder; and (4) the duplication category, which comprises disorders that are caused by deletion or duplication of contiguous genes. Our method captured a significantly low proportion (0.6%) of gene-phenotype associations in the linkage category (15.4% of the total gene-phenotype associations). During our investigation, we discovered that the linkage category includes genes whose exact locations and strands are yet to be resolved in major resources including the UCSC genome browser, Ensembl, and Gene Cards. This may explain why we detected a low number of associations in the linkage category. By contrast, factoring in the 3D genome organisation significantly increased the chance of identifying

gene-phenotype associations (i.e. from 81.9 to 98.2%) in the mutation category (Fig. 6b), which is consistent with the functional roles of the genes we identified.

## Discussion

Here we created a genetic multimorbidity atlas of traits that have the same set of genetic components (i.e. pleiotropy in spatial eQTLs or eGenes) with no particular bias to causation, confounding or endpoint effects.

The integration of spatial data enabled the identification of distal daSNP target genes that have been missed by proximity to GWAS associations. Furthermore, our method also increased the ability to detect eQTL associations that are >1 Mb apart or inter-chromosomal. It is important to note that our approach identifies only regulatory interactions that require 3D looping as

part of their mechanism of action. It remains likely that have we missed regulatory associations that occur through alternative mechanisms e.g. diffusion of regulatory factors, SNP–SNP, SNP-non-gene regulatory effects, or non-coding RNAs.

We identified greater pleiotropy in human complex diseases and phenotypes at the spatial eGene level than at daSNP level. The correlation ($r = 0.89$) between number of daSNPs and spatial eGenes suggests that phenotypes with greater number of associated SNPs are better represented in pleiotropy. Previous studies have reported pleiotropy in complex traits[44]. Our findings are consistent with the work of Sivakumaran et al.[44] in 2011, who reported 16.9 and 4.6% pleiotropy at the gene and variant levels. However, our study differs from theirs in significant ways as Sivakumaren et al.: (1) used 1687 SNPs that satisfied the GWAS significance threshold ($p$-values $< 5 \times 10^{-8}$); (2) adopted the target genes that were suggested by the authors of the GWAS, annotated in the GWAS Catalog, or are in LD with tag SNPs; and (3) reported that variant pleiotropy is associated with gene location, and that exonic variants are more pleiotropic than intergenic variants. By contrast, we used 7776 SNPs with suggestive GWAS $p$-values ($< 5 \times 10^{-6}$) and defined the target genes using spatial eQTL evidence. Notably, we find that spatial eQTLs within 1 megabase of eGenes are more than twice as common as eQTLs within genes (a in Supplementary Fig. 2). The integration of genomic organisation information into the interpretation of SNP function enabled the identification of novel regulatory interactions in complex traits. Further empirical studies are required to validate these interactions.

We hypothesise that the spatial eGene pleiotropy we identified within the phenotype clusters makes a biological contribution to the multimorbidity between the phenotypes. The most common genes in the multimorbid clusters are typically located adjacent to each other in a contiguous genomic region (Fig. 4 and Supplementary Fig. 9). We propose that these regions comprise different composite regulatory elements, each having a distinct and distinguishable effect on the genes therein, which in turn play a role in the pathogenesis of the associated complex phenotypes. The effects of these composite regulatory elements are reflected in the LD architecture of the regions. The LD data was from the 1000 Genomes CEU population (whereas the GTEx population is 85.2% Caucasian), however the results indicate that the inheritance of these regions may be linked to physical association between regions that are separated in the linear sequence. Moreover, the finding of large effect sizes for eQTLs involving variants in genomic regions with low LD is consistent with previous observations of greater deleterious effects, and larger per-SNP heritability[45] for poorly linked variants, while genomic regions with high LD have lower heritability and greater exonic deleterious effects[46].

The concentration of multiple intronic spatial eQTLs within low recombination cluster regions indicates inherited allelic heterogeneity (i.e. multiple signals at a locus that affect a trait)[47]. This is consistent with evidence that discrete multiple variants (and not a single causal variant) within an LD block impact multiple linearly separated enhancers and the expression of target genes[11,48,49]. However, causative variants cannot be separated from disease modifiers at this level because LD is subject to allele frequency, recombination, selection, genetic drift and mutation[50,51]. As such, variants in LD can affect each other's statistical values[52].

Our finding that FADS1 (along with MYRF, FEN1 and FADS3) and FADS2 (along with TMEM258) are inversely associated with eQTLs located across the FADS locus informs on the mechanism through which genetic variation contributes to the biochemistry of PUFA synthesis in complex multimorbid disorders. FADS1 and FADS2 encode the delta-5 (D5D) and delta-6 desaturase (D6D) enzymes, respectively, which catalyse the rate-limiting steps in PUFA biosynthesis respectively[32,53]. Inhibition of D6D, which acts upstream of D5D in the pathway, has been correlated with decreases in inflammation in several rodent studies[53–55]. Our results are consistent with a significant genetic (the FADS1 variant rs174548 [Fig. 4; Supplementary Fig. 8]) contribution to reduced expression of D5D, which in turn leads to a build-up of pro-inflammatory eicosanoids (via n-6 PUFA). Notably, only 5 genetic variants (rs422249, rs174448, rs174449, rs1000778 and rs174574) increase D5D transcript levels and thus favour the synthesis of anti-inflammatory eicosanoids.

TAD boundaries are generally considered to be conserved across tissues and developmental stages[56,57]. However, differences in TAD formation do occur[58,59]. We observed both intra- and inter-TAD spatial eQTL-eGene interactions, in addition to eQTLs involving variants located at TAD boundaries. For example, eQTLs rs8042374 in the CHRNA locus (about which the lung disorders cluster, Supplementary Fig. 9) and rs174537 in the FADS locus (Supplementary Fig. 7) both lie at a TAD boundary. This is consistent with observations that genetic mutations at TAD boundaries can impact on enhancer-promoter interactions[60,61]. It remains possible that cell line, developmental or cellular state-specific chromatin interactions[62,63] have been missed in the HiC libraries we used to identify the eQTL-eGene and inter-phenotype relationships due to the heterogeneity of sources of the GWAS, HiC, and GTEx data. Future work can overcome this limitation by focusing on tissue-specific Hi-C library formation to enable the teasing apart of the nuances associated with cell and tissue-specific chromatin interactions in the complex disorders of interest (e.g. using pancreatic islet for type 2 diabetes[64]).

Several studies have shown that both large-effect rare variants and small-effect variants are associated with complex diseases[49,65,66]. Yet, there is no evidence that the rare variants located at the gene locus are the main drivers of the genetic variance[67]. Our analysis of the OMIM database suggests that genes harbouring rare variants with large effects are also distally regulated by common variants with small effects.

We hypothesise that the common genes within a phenotype cluster highlight the underlying molecular mechanisms that drive shared multimorbidity (Fig. 7a). By contrast, the unique presentations of individual phenotypes within a multimorbid cluster result from molecular mechanisms driven by genes that are not shared by other members of the cluster (Fig. 7a). The genetic contribution to the regulation of multimorbidity is explained by three non-exclusive models: (1) genetic variants that are associated with more than one disease phenotype affect the same target genes (Fig. 7b), indicating genetic pleiotropy; (2) different genetic variants associated with the multimorbid phenotypes mark a single regulatory element (e.g. a super-enhancer) and thus common gene(s) (Fig. 7c); or (3) different variants each marking different regulatory elements that target the same gene (Fig. 7d).

In conclusion, the integration of spatial interaction and gene eQTL information with phenotype association data leads to the identification of the genetic components that encode the molecular mechanisms that underlie both the multimorbidity and the unique development of complex disorders and traits. Further refinement of these relationships will require empirical studies that integrate multi-omics and epigenetic information on cells and tissues from patients with multimorbid disorders.

## Methods

**Data and reference files**. The genetic variants used in this study were single-nucleotide polymorphisms (SNPs) from all traits in the GWAS Catalog (www.ebi.ac.uk/gwas; v1.0.1, downloaded on 25 August 2016) with $p$-values $< 5 \times 10^{-6}$. We defined a phenotype as the trait associated with a SNP in the GWAS Catalog. A

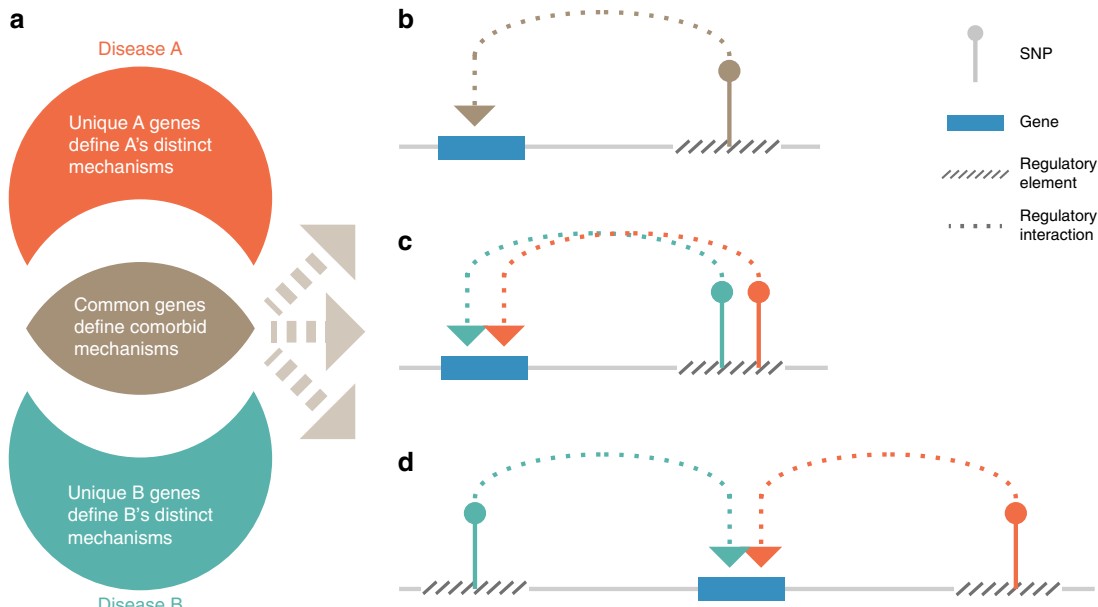

**Fig. 7** Schematic model of gene pleiotropy in multimorbidities. **a** Common target genes between any two complex disorders highlight the molecular mechanism(s) that underlie their common pathogenicity. The sets of target genes that are unique to the disorders represent the mechanisms that make the disorders different. Gene pleiotropy in multimorbidities of complex disorders can occur when **b** a variant associated with the disorders marks a regulatory element that target the common gene; **c** different variants associated with different disorders mark the same (super)-regulatory element that impacts the common gene; or **d** different regulatory elements marked by different variants impact the common gene

composite phenotype was created when more than one trait are associated with a SNP in a single study. Genomic positions of SNPs were obtained from the human hg19 genome build chromosome bed files downloaded from the NCBI (see Data Availability). We used the GENCODE transcript model (see Data Availability) as the reference for gene annotations. The GENCODE transcript model is also used in GTEx. All isoforms of a gene were collapsed into a single composite gene region. The human genome reference used in this study is the hg19 (GRChr37) build of the human genome release 75 (see Data Availability).

**Identification of spatial SNP-gene pairs.** Rao et al[20] have previously prepared high resolution Hi-C chromatin interaction libraries of the seven cell lines (i.e. GM12878, HMEC, HUVEC, IMR90, K562, KBM7 and NHEK) used in this study. In summary, they used an in-house pipeline that included BWA alignment of paired end reads against the hg19 build of the human genome. They then filtered out ambiguous alignments, difficult to align read pairs, and duplicates. We downloaded the cleaned mapping locations of Hi-C read pairs (i.e. HIC*_merged_nodups.txt files) from Gene Expression Omnibus (GEO accession number = GSE63525, *Supplementary Table 1) using the download_default_data module in CoDeS3D. The required data format is such that the rows in the interaction file describe the alignment of both read pairs (i.e. 1 and 2) with the following columns: read name, strand1, chromosome1, position1, fragment1, strand2, chromosome2, position2, fragment2. CoDeS3D works with any Hi-C pipeline that can generate interaction files in the required data format (e.g. HOMER, Juicer).

The hg19 human genome sequence was digested with the same restriction enzyme employed in preparing the Hi-C libraries (i.e. MboI) using the Restriction package of Biopython in the digest_genome module of CoDeS3D. The genome digestion produced a table of DNA fragments, which is queried by the process_inputs and find_interactions modules of CoDeS3D to identify the fragments harbouring SNPs, and their paired fragments. Next, CoDeS3D. find_genes was used with the intersect command in pybedtools to identify paired fragments that overlap with a composite gene region within the reference genome (Fig. 1a). There was no binning or padding around restriction fragments to obtain gene overlap. Genes within, or overlapping, restriction fragments, which are in contact with fragments containing the daSNPs were identified as spatial pairs to the SNP-containing fragments.

**Identification of spatial eQTL-eGene pairs.** The resulting SNP-gene pairs were then used to query the GTEx database (www.gtexportal.org, multi-tissue eQTLs analysis v4) to identify eQTL-eGene pairs i.e. SNPs that are associated with a change in the expression genes in at least one tissue. The Test Your Own API on the GTEx portal allows for the analysis of both cis and trans eQTL associations. Spatial SNP-gene pairs that have no eQTL association are excluded from the subsequent steps, which include multiple testing. The false discovery rate of the eQTL associations p-values were adjusted using the Benjamini–Hochberg

procedure, and associations with adjusted p-values < 0.05 were deemed spatial eQTL-eGene pairs.

**Construction of phenotype matrices.** A mxn matrix of $a_{i,j}$ was constructed, where m and n are the same set of phenotypes and a is the proportion of eGenes in phenotype i that are common with phenotype j. We defined a phenotype as the trait associated with a SNP in the GWAS Catalog. Sometimes more than one trait are associated with a SNP in a single study, in that case we created a composite phenotype of the traits. The pairwise ratio of common eGenes between phenotype i and phenotype j was calculated as the number of their common genes divided by the total number of eGenes associated with phenotype i,

$$\text{ratio}\, i = \frac{\text{eGenes}\, i \cap \text{eGenes}\, j}{\text{eGenes}\, i} \quad (1)$$

A similar matrix of pairwise eQTL ratios was also constructed.

To control for the eGene matrix, all 7917 eGenes were pooled together and randomly assigned to phenotypes so that each phenotype in the control matrix had the same number of eGenes as its corresponding phenotype in the eGene matrix. The pairwise ratios of common eGenes among the phenotypes were calculated as done in the eGene matrix. 1000 different null datasets were constructed in this manner and the mean matrix was calculated.

**Convex biclustering of phenotypes.** To group phenotypes based on the eGenes they share, we selected only phenotypes that have ≥4 eGenes in common. We used the cvxbuclustr R package, a convex biclustering algorithm that simultaneously groups observations and features of high-dimensional data[24]. A combined Gaussian kernel with k-nearest neighbour weights of the phenotype eGene ratios matrix was constructed. A biclustering solution path of 100 equally spaced γ parameters from $10^0$ to $10^3$ was initialised and a validation using the cobra_validate function was performed to select a regularisation parameter γ, on which the biclustering models would be based. The biclust_smooth function was used to generate a bicluster heatmap of data smoothed at the model with the minimum validation error (Uγ*).

**Multimorbidity analysis.** To identify eGenes that are central to phenotype biclusters, an eGene commonality index was calculated for each eGene in the cluster. We defined the commonality index of an eGene as the ratio of phenotypes in a cluster that are associated with that eGene. Mapping of eQTL-eGene interactions and their effects in the fat metabolism cluster was based on GTEX v7 multi-tissue analysis and hg19 genome assembly respectively. The linkage disequilibrium analysis of eQTLs in the FADS region was done on CEU population data obtained from LDLink 3.0 (https://analysistools.nci.nih.gov/LDlink/)[68]. Visualisation of Hi-

C, H3K4me1, H3K27ac, DNAse and Pol2 data in the GM12878 cell line was done with HiGlass (higlass.io)[69].

**OMIM analysis**. The 'genemap2' data was obtained from the OMIM database (omim.org, accessed 11/08/2017). Spatial eGenes that are included in the OMIM database were analysed for gene-phenotype mapping methods and compared with the OMIM genes. OMIM's phenotype-gene mapping methods are numbered thus: (1) Gene has unknown underlying defect but is associated with the disorder. (2) Disorder is mapped to gene based on linkage but mutation in gene has not been found. (3) A mutation in the gene has been identified as the basis of the mapped disorder. (4) Disorder is caused by deletion or duplication of contiguous genes.

**URLs**. For GWAS Catalog, see https://www.ebi.ac.uk/gwas/. For GTEx portal, see https://www.gtexportal.org/home/. For LDLink 3.0, see https://analysistools.nci.nih.gov/LDlink/. For HiGlass, see higlass.io.

**Code availability**. CoDeS3D pipeline is available at https://github.com/alcamerone/codes3d [https://doi.org/10.5281/zenodo.1478239]

The cvxbiclustr R package version 0.0.1 was used for convex biclustering.

All Python and R scripts used for data curation, analysis, and visualisation are available at https://github.com/Genome3d/multimorbidity-atlas [https://doi.org/10.5281/zenodo.1479964]

R version 3.3.1 and RStudio version Version 1.0.143 was used for all R scripts.

All python scripts are based on Python 3.6.6 except for CoDeS3D, which is based on Python 2.7

## Data availability

The dataset generated by the CoDeS3D pipeline that support the findings in this study are available in figshare with the identifier https://doi.org/10.17608/k6.auckland.6459728.v1. Supplementary Data 1 is available in figshare with the identifier https://doi.org/10.17608/k6.auckland.7295681. Supplementary Data 2 is available in figshare with the identifier https://doi.org/10.17608/k6.auckland.7308455. Supplementary Data 3 is available in figshare with the identifier https://doi.org/10.17608/k6.auckland.7295687. Supplementary Data 4 is available is available in figshare with the identifier https://doi.org/10.17608/k6.auckland.7295702. Supplementary Data 5 is available in figshare with the identifier https://doi.org/10.17608/k6.auckland.7295711. Supplementary Data 6 is available in figshare with the identifier https://doi.org/10.17608/k6.auckland.7295843. Source data underlying Figs. 1b, c, e, f, 4, 5a, b, 6a, b, and Supplementary Figs. 1a–d, 2a–c, 3a–c, 4, 5, 6a–d, 8a–c, 9a, b, and 10a–d are provided as a data source at figshare [https://doi.org/10.17608/k6.auckland.7308944] and are also referenced in the visualization.Rmd R Markdown file in the Github repository [https://github.com/Genome3d/multimorbidity-atlas/]. The GWAS Catalog associations (version 1.0.1) data are available at https://www.ebi.ac.uk/gwas/docs/file-downloads. The Hi-C data[20] that support the findings in this study are available from GEO with accession number, GSE63525. Accession numbers of cell replicates are given in Supplementary Table 1. Human genome build hg19 (GRChr37) was downloaded from ftp.ensembl.org/pub/release-75/fasta/homo_sapiens/. SNP annotations (human genome, build hg19) were obtained from ftp://ftp.ncbi.nih.gov/snp/organisms/human_9606_b146_GRCh37p13. Gene annotations (Transcript model from GENCODE) were downloaded from http://www.gtexportal.org/static/datasets/gtex_analysis_v6p/reference/gencode.v19.genes.v6p_model.patched_contigs.gtf.gz

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

## Acknowledgements

This work was funded by High Value Nutrition National Science (MBIE/HVN grant #3710040) grant. J.M.O. and W.S. are funded by a Royal Society of New Zealand Marsden Fund (Grant 16-UOO-072).

## Author contributions

T.F. ran analyses, wrote software, interpreted data and wrote the manuscript. W.S. contributed to data interpretation and commented on the manuscript. T.L. contributed to data interpretation and commented on the manuscript. J.M.O. directed the study, contributed to data interpretation and co-wrote the manuscript. J.M.O. is guarantor for this article.

## Additional information

**Competing interests:** The authors declare no competing interests.

