## [peer review file · Nature Communications]

Reviewer #2 (Remarks to the Author):

The manuscript “Chromatin interactions and expression quantitative trait loci reveal genetic drivers of multimorbidities” from Fadason and colleagues describes findings and identifies genes that are in common across phenotypes, and those that contribute to unique phenotypes, using a novel way of combining Hi-C, eQTLs, and GWAS data.

This paper contains novel biological insights into the mechanism of common genetic basis of multimorbidities. The results, and possibly the analysis pipeline, is potentially of great interest to the scientific community.

Major comments

My main comment is that some information on the method is missing (selection of SNP-gene targets before using convex biclustering) to evaluate the usefulness of the pipeline and fully understand the observations and conclusions that follow from it.

In particular, it is hard to evaluate how much the spatial (Hi-C) information adds to the functional evidence (i.e. GTEx), without greater information about how the HiC SNP-target genes were chosen compared to eQTL-target gene. What is the advantage of using HiC data to select these? How much info are we gaining compared to only using eQTL info?

Line 64 and Supplementary Fig 2: although the authors refer to “the CoDeS3D pipeline”, more detail should be added here and under Methods to help the reader give an idea of the pipeline for the SNP-target gene selection from Hi-C interaction data.

Do the interactions have an FDR/pvalue threshold? I am particularly worried about the 19.3% that are trans interactions. How many of these are inter-chromosomal results? These are normally discarded since believed to be false positives in interaction data.

Line 67: what is the distribution of the Hi-C loops for the 1,183,037 SNP-gene pairs analyzed, i.e. mean distance, are most distributed nearby? Add histogram of loop distance or plot similar to Supp Fig 2 for Hi-C. How many of the pairs target multiple genes? Add boxplot of SNP-pairs targeting multiple genes (e.g. 1, 2, 3, 4, 5, >5).

How would the results of this analysis change if only the eQTL data was used, without the Hi-C data, to select disease SNP-gene pairs? Would it be possible to repeat the same analysis using all SNP-gene pairs in GTEx filtered by FDR? If this includes too many pairs, would it be possible to do this for one chromosome only, or at least give an idea of the pairs and results if the analysis was run using only information from eQTLs?

Lines 134-135 about common eGenes lying in contiguous genomic region: can we get an intuition about this important biological finding? Is it driven by how the interacting regions of target genes were defined (e.g. binning, padding around restriction fragment, etc)? Has this been observed in other studies, using only eQTL data? It would be interesting to see if this is true also if biclustering was used on eQTLs without the interactions, since interactions define regions while eQTLs link SNPs to eGenes.

Line 620 and Supplementary Figure 2: Another important finding is described here (that there are more eQTLs within 1 mb of genes than there are within genes). Again (as above) it would be good to reassure that this is not an artifact from selection of the pairs using the interaction data.

Lines 204-217: Is it possible to show the pairwise correlation between effect sizes in a similar way as the correlation between SNPs (LD) is shown, for the top eQTLs in this region (Fig S6)? Otherwise a figure that illustrates directly how LD is related to correlation/direction of effects?

Lines 210-214: What is the frequency of these variants? Please make sure this is not driven by errors in effect allele flipping.

Discussion on greater gene pleiotropy vs. SNP pleiotropy: I am not sure if you can make this statement in general since this evaluation is limited to SNPs that are connected to disease through genes. Maybe there are more complex mechanisms for SNP pleiotropy if considering all GWAS SNPs overlapping Hi-C (not limited to the connections overlapping a gene).

Minor comments

Line 15-16 of the abstract mentions “metabolic capacity”, seems out of context without further explanation and would remove.

Line 34-36: First line of Intro also can be more specific to the goals in this paper, or explain better how understanding genes behind multimorbidities leads to personalized medicine and assessing patient’s overall health.

Lines 71-72 and S1 Figure: mention why chromosome 22 is missing, chromosome 22 does not have any daSNP, or GTEx does not contain any significant eQTLs on chromosome 22?

Line 84: “were correctly mapped in the GWAS Catalogue”, assuming eQTL is the correct mapping. Maybe better to say that these were mapped to the same gene as the eQTL?

What is Supplementary Fig 1f? (not mentioned in the main text)

Fig S4 – did you mean shared eQTLs and not shared eGenes?

Line 121-122: how many phenotypes considered in biclustering that share ≥ 4 genes? 615 or 618?

Line 130, add eQTL.

Reorder Supp figures S6 and S9 (CHRNA example after)

Lines 207 and Figure 4a. What is the asterisk on top of categories E, B, D mean? Is this from the mean effect size across all tissues, or only significant eQTLs?

Figure 4 and line 301: It is hard to digest the meaning of many categories with respect to LD, if the important finding is opposite effects on common genes, is it possible to just collapse the categories?

Reviewer #3 (Remarks to the Author):

In this manuscript, Fadason and colleagues use convex biclustering to identify phenotype clusters identified genes that show physical interaction through long range looping events identified by HiC versus eQTL gene correlations. The approach parallels closely a similar report from this team published in 2017 focusing on Type 2 diabetes and obesity genes but is now expanded to include the whole GWAS database. The important findings here relate to the power of using spatial relationships in defining the genes over simple eQTLs to observe co-occurrence of morbid phenotypes. Overall this is an interesting analysis and improves upon previous approaches exploring pleiotropy in complex genetic diseases. The following issues should be addressed:

- 1) This analysis is limited in that the spatial interactions are derived from HiC data using immortalized cell lines while the eQTL expression data is taken from GeTex which is largely primary tissues. It is expected that the spatial data will be limited in its ability to represent the true 3D interactions in the GeTex samples. This is mentioned in the discussion but it could be stated more clearly.
- 2) The associations between comorbidity eGene clusters and the disease phenotypes is expected in part since this information is why they are in the GWAS catalog in the first place. More attention in the writing could be given to emphasize what is really new in this analysis in relation to what is already known from the GWAS data. For instance, how many eGenes are the same as the genes associated with eQTLs and how many would not have been considered without spatial data?
- 3) Is multimorbidity the same as pleiotropy in this analysis, please clarify?
- 4) In the first paragraph of the results section, what does “unique” snps mean here exactly? Are these SNP all independent by LD? If these SNPs are in LD how many unique haplotypes does this define?
- 5) In the last paragraph of the first results section on page 5, the description of trans effects is confusing since in the paragraph above it is mentions that 41 genes on ChrX have trans eQTL effects from other chromosomes but here, trans effects are defined not by different chromosome but by arbitrary distance on the same chromosome. Moverover, for ChrX what is the power for convincing identification of trans-eQTL effects?
- 6) The results section on multimorbid phenotypes cluster around shared egenes, demonstrates that spatial information adds important relational data to the analysis however, it would be interesting to know what the relationship would look like with genes demonstrating association with eQTLs, the proximal genes. If this is what the eQTL analysis is (a in S3) then it needs clarification that these are eQTL genes and not the actual variants. If not, then the associated eQTL genes would be an important control for the added value of the spatial data.

- 7) At the end of the first paragraph on page 7, It is unclear why this number of SNPs is chosen for comparison. eGenes and SNPs are not directly comparable based on integer alone. Was this derived based on testing a range of SNPs?
- 8) In the second paragraph on page 7 the authors state that it is noteworthy that eGenes lie adjacent to each other in a contiguous genomic region. Can the authors offer some reason why they think this is noteworthy rather than simply describing the observation?
- 9) On page 11, referencing figure 3B, we are asked to compare this to figure S6c. It is not clear from the text what we are supposed to compare between these two figures. Some simplification here would be helpful to the reader.
- 10) On page 13, in the discussion of the CHRNA locus, it would be interesting to see how these data looked when LD is defined by D' , whereby linked SNPs with different allele frequencies might show more of these effects. Also, it would be expected that eQTLs would also demonstrate gene and tissue specific patterns so how does the eGene information change or add to this?
- 11) The section on GWAS eQTLs and medelian genes and the accompanying figure needs more explanation in the text. It is not clear how the conclusions are drawn.
- 12) In addition, some more detail about the results and analysis added to the text and some simplification of figures 3, S6 and S7 in the text would make the points addressed easier to comprehend.

Dear Editor,

Thank you and the reviewers for some very positive and insightful comments into our manuscript. As a result of your comments, we have performed some additional analyses and rewritten some sections of the manuscript. We strongly believe that this revision has significantly improved our study and manuscript. We have provided a detailed list of our changes below. We look forward to hearing from you.

Sincerely

Justin M. O'Sullivan

Reviewer #2

1) My main comment is that some information on the method is missing (selection of SNP-gene targets before using convex biclustering) to evaluate the usefulness of the pipeline and fully understand the observations and conclusions that follow from it.

⇒ We have modified the *Identification of spatial eQTL-eGene pairs* section of Methods to correct this omission. The section now reads:

...The hg19 human genome sequence is digested with the same restriction enzyme used in the preparing the HiC libraries (*i.e.* MboI) using the Restriction package of Biopython. Genes within restriction fragments which are in contact with fragments containing the daSNPs were identified... Spatial SNP-gene pairs that have no eQTL association are excluded from the subsequent steps, which include multiple testing. The false discovery rate of the eQTL associations p-values were adjusted using the Benjamini-Hochberg procedure, and associations with adjusted p-values < 0.05 were deemed spatial eQTL-eGene pairs.

2) In particular, it is hard to evaluate how much the spatial (Hi-C) information adds to the functional evidence (*i.e.* GTEx), without greater information about how the HiC SNP-target genes were chosen compared to eQTL-target gene. What is the advantage of using HiC data to select these? How much info are we gaining compared to only using eQTL info?

⇒ We agree that this information should have been included in the original manuscript and thank the reviewer for asking this question. We analysed the set of daSNPs located on chromosome 22 using CoDeS3D and a standard cis-eQTL analysis in GTEx. This enabled us to compare the relative information gained by including the spatial (*i.e.* HiC data) into the analysis. We have included this information in S3 Fig and added the following text to the *GWAS SNPs mark spatial regulatory regions* section of the **Results**:

“To estimate how much functional information is gained by the integration of spatial data, we took all 339 daSNPs on chromosome 22 and queried the GTEx v7 analysis for significant eQTL associations. The ‘GTEx only’ method yielded 4,408 eQTL associations, all of which were within 1 Mb genomic distance (a and b in S3 Fig). We then analysed the same set of 339 daSNPs using the CoDeS3D pipeline, which integrates GTEx v7 analysis. The CoDeS3D approach identified 4,543 spatial eQTL associations of which 3,542 (~78%) were also found in the ‘GTEx only’ associations (a and b in S3 Fig). 866 (19.6%) of the ‘GTEx only’ associations, with eQTL effect of

-1.11 to 1.15, were lost in the CoDeS3D analysis due to lack of evidence for a spatial connection between the genomic fragments containing the SNP and gene. Conversely, the integration of spatial data enabled the detection of 1,001 eQTL associations, with eQTL effect sizes ranging between -1.61 and 0.74, that were not detected by the ‘GTEx only’ method (a and c in S3 Fig). These results are consistent with a significant number of eQTL effects involving 3D looping interactions, up to and beyond 1 Mb. The absence of direct physical contacts for all eQTLs is consistent with alternative mechanisms, including the diffusion of regulatory factors released from the eQTL locus (Dekker & Mirny, 2016), contributing to the regulatory network.”

⇒ We have also included a discussion of the implications of adding the spatial component to the eQTL analysis in the second paragraph of the **Discussion**:

“...Furthermore, our method also increased the ability to detect eQTL associations that are > 1 Mb apart or inter-chromosomal. It is important to note that our approach identifies only regulatory interactions that require 3D looping as part of their mechanism of action. It remains likely that we have missed regulatory associations that occur through alternative mechanisms *e.g.* diffusion of regulatory factors or non-coding RNAs.”

3) Line 64 and Supplementary Fig 2: although the authors refer to “the CoDeS3D pipeline”, more detail should be added here and under Methods to help the reader give an idea of the pipeline for the SNP-target gene selection from Hi-C interaction data.

⇒ We have modified the methods section - *Identification of spatial eQTL-eGene pairs*- to outline the CoDeS3D pipeline more clearly.

4) Do the interactions have an FDR/pvalue threshold? I am particularly worried about the 19.3% that are trans interactions. How many of these are inter-chromosomal results? These are normally discarded since believed to be false positives in interaction data.

⇒ In our analysis, we did not exclude any spatial SNP-gene pair until after the eQTL analysis, and then only if there was no eQTL association. Filtering was completed by selecting for an FDR of <0.05 following a Benjamini-Hochberg calculation for each eQTL-eGene pair that was identified. We contend that adjusting p values after the integration of the spatial component is a Bayesian-like step in which the probability of the eQTL association is strengthened given the evidence of a spatial interaction across cell line replicates. In total 1,473 out of 7,917 eGenes are affected by inter-chromosomal eQTLs. While we agree that the number is high, we also note that functional inter-chromosomal interactions have been previously reported *e.g.* in MCF breast cancer genome (Barutcu et al., 2015), and in peripheral blood (Lukowski et al., 2017).

5) Line 67: what is the distribution of the Hi-C loops for the 1,183,037 SNP-gene pairs analyzed, i.e. mean distance, are most distributed nearby? Add histogram of loop distance or plot similar to Supp Fig 2 for Hi-C. How many of the pairs target multiple genes? Add boxplot of SNP-pairs targeting multiple genes (*e.g.* 1, 2, 3, 4, 5, >5).

⇒ We have inserted the required information into the manuscript. In brief, 16,248 out of the 1,183,037 SNP-gene pairs that were analysed had eQTL associations. We have modified the *GWAS SNPs mark spatial regulatory regions* result section to include: 1) the target

gene distribution of each eQTL SNP; and 2) the Hi-C loop distances for 16,248 eQTL-eGene pairs.. The text now reads:

“...57.8% (4,577) of eGenes are affected by only one eQTL SNP, while two eGenes are affected by > 32 eQTL SNPs (f in S1 Fig). On the other hand, ~49% of eQTL SNPs affect more than one gene (b in S2 Fig). The eQTL SNP-gene Hi-C fragment loop distances range from 0 bp to 248 Mb, with a median and mean values of 13,680 bp and 575,100 bp respectively (c in S2 Fig).”

6) How would the results of this analysis change if only the eQTL data was used, without the Hi-C data, to select disease SNP-gene pairs? Would it be possible to repeat the same analysis using all SNP-gene pairs in GTEx filtered by FDR? If this includes too many pairs, would it be possible to do this for one chromosome only, or at least give an idea of the pairs and results if the analysis was run using only information from eQTLs?

⇒ We have addressed this by analyzing daSNPs on chromosome 22, as detailed in our response to point 2) above.

7) Lines 134-135 about common eGenes lying in contiguous genomic region: can we get an intuition about this important biological finding? Is it driven by how the interacting regions of target genes were defined (e.g. binning, padding around restriction fragment, etc)? Has this been observed in other studies, using only eQTL data? It would be interesting to see if this is true also if biclustering was used on eQTLs without the interactions, since interactions define regions while eQTLs link SNPs to eGenes.

⇒ The spatial interactions between eQTL SNPs and their target genes are defined according to captured interactions between restriction fragments. We use raw HiC interaction files and there was no binning or padding around restriction fragments.

⇒ We are not aware of any study that has reported common eGenes lying in contiguous genomic regions on the basis of eQTL data.

⇒ Given that the contiguous genomic regions are typically 100kb – 400kb in length, the biclustering of the eGenes associated with eQTLs (as analysed by GTEx) without the spatial component should identify some of the same contiguous regions. However, the way we have performed the biclustering of spatial interactions informs on: 1) the common contiguous region; and 2) what is unique among the phenotypes.

⇒ We have elaborated our answer to this query within the *Mutimorbid phenotypes cluster around shared eGenes* section of the **Results**:

“Multiple variants from one genomic region have been associated with cross-phenotypes before *e.g.* the *IFI30* locus in autoimmune diseases(Farh et al., 2015), and the *CDKN2B-ASI* locus in coronary artery disease, glioma, and intracranial aneurysm(Solovieff, Cotsapas, Lee, Purcell, & Smoller, 2013). However, these studies did not resolve the target genes of the variants. Alteration of the dosages of multiple adjacent contiguous genes by copy number variants (CNVs) have also been associated with the pathogenicity of contiguous gene syndromes (*e.g.* Prader-Willi syndrome, Angelman syndrome and Williams syndrome) (Pereira & Marion, 2018; Schmickel, 1986). These dosage-sensitive genes in contiguous gene syndromes are typically placed in genomic regions of <5 Mb and contribute to phenotypes

independently (Shaffer, Ledbetter, & Lupski, n.d.). Therefore, it is noteworthy that eGenes that are common to most phenotypes in the clusters we detected lie adjacent to each other in a contiguous genomic region (typically 100 - 400 kb in length) and are in cis-association with the eQTL SNPs. This is exemplified by: a subset of immune related disorders that cluster about the *PGAP3* – *GSDMA* locus (Chr 17: 37,827,375 – 38,134,431; hg19); skin pigmentation and skin cancer which are clustered about the *SPATA33* – *URAHP* region (Chr 16:89,724,152 – 90,114,191; hg19); and a mood disorder cluster that is built about the *NT5DC2* - *TMEM110* locus (Chr 3:52558385 – 52931597; hg19; S6 Fig). This study, to the best of our knowledge, is the first to observe contiguous target genes of eQTL SNPs in complex cross-phenotypes.”

8) Line 620 and Supplementary Figure 2: Another important finding is described here (that there are more eQTLs within 1 mb of genes than there are within genes). Again (as above) it would be good to reassure that this this is not an artifact from selection of the pairs using the interaction data.

⇒ As discussed above (point 7), there is no pre-selection of the pairs for the interaction data. Rather we are using HiC data at the restriction fragment level.

9) Lines 204-217: Is it possible to show the pairwise correlation between effect sizes in a similar way as the correlation between SNPs (LD) is shown, for the top eQTLs in this region (Fig S6)? Otherwise a figure that illustrates directly how LD is related to correlation/direction of effects?

⇒ We thank the reviewer for their suggestion. We have modified Fig 4 to include a visualization of this and believe it makes a strikingly informative figure.

10) Lines 210-214: What is the frequency of these variants? Please make sure this is not driven by errors in effect allele flipping.

⇒ Allele flipping is a possible contributor to some of the reversals in regulation we observe. We have presented the different variant frequencies in Table S4. We have also inserted the following text in the results section:

“It is possible that the opposite regulatory effect observed for these five eQTLs represents allele flipping (S4 Table).”

11) Discussion on greater gene pleiotropy vs. SNP pleiotropy: I am not sure if you can make this statement in general since this evaluation is limited to SNPs that are connected to disease through genes. Maybe there are more complex mechanisms for SNP pleiotropy if considering all GWAS SNPs overlapping Hi-C (not limited to the connections overlapping a gene).

⇒ We agree with the reviewer. The methodology we have proposed does have limitations – as we have previously outlined (Fadason et al. 2017). In particular we do not identify or characterize the contribution of SNP-SNP or SNP-non-gene interactions due to the lack of suitable functional data (e.g. eQTL data requires expression). Therefore, we have inserted the following qualifier in the **Discussion**:

The integration of spatial data enabled the identification of distal daSNP target genes that have been missed in the GWAS associations. Furthermore, our method also increased the ability to detect eQTL associations that are > 1 Mb apart or inter-chromosomal. It is important to note that our approach identifies only regulatory interactions that require 3D looping as part of their mechanism of action. It remains likely that we have missed regulatory associations that occur through alternative mechanisms *e.g.* diffusion of regulatory factors, SNP-SNP, SNP-non-gene, or non-coding RNAs.

Minor

comments

Line 15-16 of the abstract mentions “metabolic capacity”, seems out of context without further explanation and would remove.

⇒ We have modified the text to read: “Clinical studies of non-communicable diseases identify multimorbidities that suggest a common set of predisposing factors.”

Line 34-36: First line of Intro also can be more specific to the goals in this paper, or explain better how understanding genes behind multimorbidities leads to personalized medicine and assessing patient’s overall health.

⇒ We thank the reviewer for this suggestion and have inserted the following into the second line of the introduction: “At the same time, there is great expectation that personalised medicine will aid in delivering medical care that is more suitable to the individual.”

Lines 71-72 and S1 Figure: mention why chromosome 22 is missing, chromosome 22 does not have any daSNP, or GTEx does not contain any significant eQTLs on chromosome 22?

⇒ We have fixed this error. We have also inserted the following text in the legend to S1 Fig: “For c) and d); X axis tick marks are organized sequentially from 1-22, followed by the X chromosome. The Y chromosome is not represented on c) or d).”

Line 84: “were correctly mapped in the GWAS Catalogue”, assuming eQTL is the correct mapping. Maybe better to say that these were mapped to the same gene as the eQTL?

⇒ We agree and have modified the text to read: “Only 24.3% of the eQTL-eGene pairs matched the SNP-gene mapping in the GWAS Catalogue...”

What is Supplementary Fig 1f? (not mentioned in the main text)

⇒ Thank you for pointing out this omission. We have now mentioned Fig. 1f in-text in paragraph 3 of the *GWAS SNPs mark spatial regulatory regions* section of **Results**.

Fig S4 – did you mean shared eQTLs and not shared eGenes?

⇒ Thank you. We meant shared eQTLs. We have corrected this.

Line 121-122: how many phenotypes considered in biclustering that share ≥ 4 genes? 615 or 618?

⇒ We have corrected this. It is 618.

Line 130, add eQTL.

⇒ Done, thank you.

Reorder Supp figures S6 and S9 (CHRNA example after)

⇒ We have reordered the figures. Thank you.

Lines 207 and Figure 4a. What is the asterisk on top of categories E, B, D mean? Is this from the mean effect size across all tissues, or only significant eQTLs?

⇒ We have completely redrawn Figure 4 and clarified this on the reviewer's suggestion. We believe it is much more self-explanatory now.

Figure 4 and line 301: It is hard to digest the meaning of many categories with respect to LD, if the important finding is opposite effects on common genes, is it possible to just collapse the categories?

⇒ See above.

Reviewer #3 (Remarks to the Author):

1) This analysis is limited in that the spatial interactions are derived from HiC data using immortalized cell lines while the eQTL expression data is taken from GeTex which is largely primary tissues. It is expected that the spatial data will be limited in its ability to represent the true 3D interactions in the GeTex samples. This is mentioned in the discussion but it could be stated more clearly.

⇒ We agree with the reviewer and have modified paragraph seven in the discussion to reflect this:

It remains possible that cell line, developmental or cellular state specific chromatin interactions (Fraser, Williamson, Bickmore, & Dostie, 2015; Fuchsberger et al., 2016) have been missed in the HiC libraries we used to identify the eQTL-eGene and inter-phenotype relationships due to the heterogeneity of sources of the GWAS, HiC, and GTEx data. Future work can overcome this limitation by focusing on tissue specific Hi-C library formation to enable the teasing apart of the nuances associated with cell and tissue specific chromatin interactions in the complex disorders of interest...

2) The associations between comorbidity eGene clusters and the disease phenotypes is expected in part since this information is why they are in the GWAS catalog in the first place. More attention in the writing could be given to emphasize what is really new in this analysis in relation to what is already known from the GWAS data. For instance, how many eGenes are the same as the genes associated with eQTLs and how many would not have been considered without spatial data?

⇒ We have addressed this concern by clarifying the information within the manuscript. This includes the addition of the following:

Only 24.3% of the eQTL-eGene pairs matched the SNP-gene mapping in the GWAS Catalogue and these contribute 18.5% of the tissue specific eQTL-eGene interactions (e in S1 Fig). 13,240 (75.7%) eQTL-eGene pairs are missed in the GWAS mapping of genes to SNPs. (Paragraph 3 in the *GWAS SNPs mark spatial regulatory regions* section of **Results**.)

3) Is multimorbidity the same as pleiotropy in this analysis, please clarify?

⇒ We agree this is an important point. We have changed the discussion as follows:

“We hypothesise that the gene pleiotropy we identified within the phenotype clusters makes a biological contribution to the multimorbidity between the phenotypes.”

4) In the first paragraph of the results section, what does “unique” snps mean here exactly? Are these SNP all independent by LD? If these SNPs are in LD how many unique haplotypes does this define?

⇒ We thank the reviewer for pointing this out. We have removed the term ‘unique’ from the text. The LD between the SNPs becomes an interesting feature of the regulatory networks that we observe (see Fig 4 and S9 a and b).

5) In the last paragraph of the first results section on page 5, the description of trans effects is confusing since in the paragraph above it is mentions that 41 genes on ChrX have trans eQTL effects from other chromosomes but here, trans effects are defined not by different chromosome but by arbitrary distance on the same chromosome. Moreover, for ChrX what is the power for convincing identification of trans-eQTL effects?

⇒ We agree this needs clarification. We have clarified the definition of the cis and trans effects in the text:

“... were associated with only *cis*-spatial interactions (*i.e.* both partners are from the same chromosome and separated by <1,000,000 bp, as defined elsewhere(Ardlie et al., 2015; Taneera et al., 2012)), 19.3% (1,528) by only *trans*-interactions (*i.e.* both partners are from the same chromosome and separated by >1,000,000 bp, or from different chromosomes (Ardlie et al., 2015; Taneera et al., 2012)).

⇒ We included citations 24 and 25 to justify the decision to limit cis-effects to loci separated by <1,000,000 bp.

⇒ We have also modified the text that refers to the connections on the x-chromosome specifically. It now reads:

“However, 41 genes on the X chromosome are affected by eQTLs from other chromosomes”

⇒ The trans-eQTLs that were affecting the eGenes on the ChrX were all significant with adjusted p values less than the FDR of 0.05.

6) The results section on multimorbid phenotypes cluster around shared egenes, demonstrates that spatial information adds important relational data to the analysis however, it would be interesting to know what the relationship would look like with genes demonstrating association with eQTLs, the proximal genes. If this is what the eQTL analysis is (a in S3) then it needs clarification that these are eQTL genes and not the actual variants. If not, then the associated eQTL genes would be an important control for the added value of the spatial data.

⇒ Biclustering analysis using the nearest gene results in an extremely fragmented cluster (see below). We contend that the tests that we have shown (S4 and S5 Fig) provide the reader with sufficient and appropriate controls. We believe that inclusion of the proximal gene bicluster graph will create confusion for the reader (given the altered orders of phenotypes and reduced clustering that is observed).

detected lie adjacent to each other in a contiguous genomic region (typically 100 - 400 kb in length) and are in cis-association with the eQTL SNPs. This is exemplified by: a subset of immune related disorders that cluster about the *PGAP3* – *GSDMA* locus (Chr 17: 37,827,375 – 38,134,431; hg19); skin pigmentation and skin cancer which are clustered about the *SPATA33* – *URAHP* region (Chr 16:89,724,152 – 90,114,191; hg19); and a mood disorder cluster that is built about the *NT5DC2* - *TMEM110* locus (Chr 3:52558385 – 52931597; hg19; S6 Fig). This study, to the best of our knowledge, is the first to observe contiguous target genes of eQTL SNPs in complex cross-phenotypes.”

⇒ We have also modified the discussion to read:

“The most common genes in the multimorbid clusters are typically located adjacent to each other in a contiguous genomic region (Fig 4 and S9 Fig). We propose that these regions are comprised of different composite regulatory elements, each having a distinct and distinguishable effect on the genes therein, which in turn play a role in the pathogenesis of the associated complex phenotypes. The effects of these composite regulatory elements are reflected in the LD architecture of the regions and indicates that the inheritance of these regions may be linked to physical association between regions that are separated in the linear sequence. Moreover, the finding of large effect sizes for eQTLs involving variants in genomic regions with low LD is consistent with previous observations of greater deleterious effects, and larger per-SNP heritability (Gazal et al., 2017) for poorly linked variants, while genomic regions with high LD have lower heritability and greater exonic deleterious effects (Hussin et al., 2015). “

9) On page 11, referencing figure 3B, we are asked to compare this to figure S6c. It is not clear from the text what we are supposed to compare between these two figures. Some simplification here would be helpful to the reader.

⇒ We have modified text to clarify this:

“We also observed this one-to-many SNP-eGene eQTL association in the pulmonary cluster, including inter-TAD connections from the region marked by rs8042374 (c in S9 Fig).”

10) On page 13, in the discussion of the *CHRNA* locus, it would be interesting to see how these data looked when LD is defined by D' , whereby linked SNPs with different allele frequencies might show more of these effects. Also, it would be expected that eQTLs would also demonstrate gene and tissue specific patterns so how does the eGene information change or add to this?

⇒ Thank you for your comment. S8 and S9 Fig now show LD data for both R^2 and D' . This comparison shows that:

“the tissue eQTL effect patterns of the linked SNPs seem to be consistent with their differences in allele frequency, as informed by the R^2 and D' scores...”

(Paragraph 1 in *eQTLs have gene and tissue specific patterns* section of **Results**).

⇒ We may have misunderstood but the tissue specific regulatory effects (e.g. shown in S9 b) represent the effect of the eQTL on the eGene that is affected by the daSNP.

11) The section on GWAS eQTLs and median genes and the accompanying figure needs more explanation in the text. It is not clear how the conclusions are drawn.

⇒ We have re-written this section. It now reads:

“There are four gene-phenotype mapping categories in the OMIM database: 1) The “unknown defect” category, in which a disorder is associated with a gene with no known functional role; 2) The “linkage” category, which maps a gene to a disorder based on linkage despite no mutation having been found in the gene yet; 3) The “mutation” category, in which a mutation in the gene has been identified as the basis of the mapped disorder; and 4) The “duplication” category, which comprises disorders that are caused by deletion or duplication of contiguous genes. Our method captured a significantly low proportion (0.6%) of gene-phenotype associations in the “linkage” category (15.4% of the total gene-phenotype associations). During our investigation, we discovered that the “linkage” category includes genes whose exact locations and strands are yet to be resolved in major resources including the UCSC genome browser and Gene Cards. This may explain why we detected a low number of associations in the “linkage” category. By contrast, factoring in the 3D genome organization significantly increased the chance of identifying gene-phenotype associations (*i.e.* 81.9% to 98.2%) in the “mutation” category (Fig 6b), which is consistent with the functional roles of the genes we identified. “

12) In addition, some more detail about the results and analysis added to the text and some simplification of figures 3, S6 and S7 in the text would make the points addressed easier to comprehend.

⇒ We thank the reviewer for raising this point. We have redrawn several figures in the manuscript to aid clarity. Figures S6 and S7 have been merged to try and clarify the important features.

References

- Ardlie, K. G., Deluca, D. S., Segre, A. V., Sullivan, T. J., Young, T. R., Gelfand, E. T., ... Dermitzakis, E. T. (2015). The Genotype-Tissue Expression (GTEx) pilot analysis: Multitissue gene regulation in humans. *Science*, *348*(6235), 648–660. <http://doi.org/10.1126/science.1262110>
- Barutcu, A. R., Lajoie, B. R., McCord, R. P., Tye, C. E., Hong, D., Messier, T. L., ... Stein, G. S. (2015). Chromatin interaction analysis reveals changes in small chromosome and telomere clustering between epithelial and breast cancer cells. *Genome Biology*, *16*(1), 214. <http://doi.org/10.1186/s13059-015-0768-0>
- Dekker, J., & Mirny, L. (2016). The 3D Genome as Moderator of Chromosomal Communication. *Cell*, *164*(6), 1110–1121. <http://doi.org/10.1016/j.cell.2016.02.007>
- Farh, K. K., Marson, A., Zhu, J., Kleinewietfeld, M., Housley, W. J., Beik, S., ... Bernstein, B. E. (2015). Genetic and epigenetic fine mapping of causal autoimmune disease variants. *Nature*, *518*(7539), 337–343. <http://doi.org/10.1038/nature13835>
- Fraser, J., Williamson, I., Bickmore, W. A., & Dostie, J. (2015). An Overview of Genome Organization and How We Got There: from FISH to Hi-C. *Microbiology and Molecular Biology Reviews*, *79*(3), 347–372. <http://doi.org/10.1128/MMBR.00006-15>
- Fuchsberger, C., Flannick, J., Teslovich, T. M., Mahajan, A., Agarwala, V., Gaulton, K. J., ... McCarthy, M. I. (2016). The genetic architecture of type 2 diabetes. *Nature*, *536*(7614), 41–7. <http://doi.org/10.1038/nature18642>
- Gazal, S., Finucane, H. K., Furlotte, N. A., Loh, P.-R., Palamara, P. F., Liu, X., ... Price, A. L. (2017). Linkage disequilibrium-dependent architecture of human complex traits shows action of negative selection. *Nature Genetics*, *49*(10), 1421–1427. <http://doi.org/10.1038/ng.3954>
- Hussin, J. G., Hodgkinson, A., Idaghdour, Y., Grenier, J.-C., Goulet, J.-P., Gbeha, E., ... Awadalla, P. (2015). Recombination affects accumulation of damaging and disease-associated mutations in human populations. *Nature Genetics*, *47*(4), 400–404. <http://doi.org/10.1038/ng.3216>
- Lukowski, S. W., Lloyd-Jones, L. R., Holloway, A., Kirsten, H., Hemani, G., Yang, J., ... Powell, J. E. (2017). Genetic correlations reveal the shared genetic architecture of transcription in human peripheral blood. *Nature Communications*, *8*(1), 483. <http://doi.org/10.1038/s41467-017-00473-z>
- Pereira, E., & Marion, R. (2018). Contiguous Gene Syndromes. *Pediatrics in Review*, *39*(1), 46–49. <http://doi.org/10.1542/pir.2016-0073>
- Schmickel, R. D. (1986). Contiguous gene syndromes: A component of recognizable syndromes. *The Journal of Pediatrics*, *109*(2), 231–241.
- Shaffer, L. G., Ledbetter, D. H., & Lupski, J. R. (n.d.). Molecular Cytogenetics of Contiguous Gene Syndromes: Mechanisms and Consequences of Gene Dosage Imbalance. In D. Valle, A. L. Beaudet, B. Vogelstein, K. W. Kinzler, S. E. Antonarakis, A. Ballabio, ... G. Mitchell (Eds.), *The Online Metabolic and Molecular Bases of Inherited Disease*. McGraw-Hill Medical.
- Solovieff, N., Cotsapas, C., Lee, P. H., Purcell, S. M., & Smoller, J. W. (2013). Pleiotropy in complex traits: challenges and strategies. *Nature Reviews Genetics*, *14*(7), 483–495. <http://doi.org/10.1038/nrg3461>
- Taneera, J., Lang, S., Sharma, A., Fadista, J., Zhou, Y., Ahlqvist, E., ... Groop, L. (2012). A systems genetics approach identifies genes and pathways for type 2 diabetes in human islets. *Cell Metabolism*, *16*(1), 122–134. <http://doi.org/10.1016/j.cmet.2012.06.006>

Reviewer #2 (Remarks to the Author):

The manuscript "Chromatin interactions and expression quantitative trait loci reveal genetic drivers of multimorbidities" contains important insight on the value of using Hi-C interaction data together with eQTL and disease/trait-associated SNPs to identify genes in complex diseases. These observations are very valuable and timely.

The authors have thoroughly addressed most of the questions from the previous reviewers, although a few need more attention along with some additional ones outlined below.

More information is needed for this analysis to be reproducible.

The following questions would be clarified if the pipeline to find loops in CoDeS3D pipeline was laid out clearly.

Please be more specific about the initial identification of the spatial eGenes step-by-step:

- Which pipelines were used in the CoDeS3D pipeline to get the raw interaction HiC data (e.g. HiC-Pro)?

- how were the HiC interaction data intersected with "genes" (please define "genes", is it from promoter regions with padded number of bases? or using the GTEx definition?)

- in Methods specify that you are using the GTEx results from both the cis and trans eQTLs

- The authors say that they used unfiltered raw interaction files, using no binning or padding. But isn't the raw interaction data from most pipelines defined at particular resolutions?

- Also, it seems as the authors used all raw interactions from Hi-C with no cutoff for FDR or number of reads, so some of these loops could be supported by only one read, is that correct? I understand that the loops are then filtered using the eQTL dataset. However, loops supported by only one read have a very high chance of being false positives. It would be good to check whether the loops with lower number of reads lack support from eQTLs and are therefore filtered out in any case.

Abstract/intro: Some inconsistency in wording could be solved by defining in the Introduction a spatial eQTL-eGene pair as a daSNP-gene pair which is supported by both interaction and eQTL data, and using the same working throughout.

Figure 1: are the phenotypes with more number of reported daSNPs better represented? If this is the case, please specify in Discussion for caveats.

Line 96: out of the 1,528 trans-interactions, how many are on different chromosomes? Please specify the number for these here.

Lines 101-102: can remove since it is already in Supp Fig 2C.

Line 114 and Line 284: how were the strength of eQTL associations measured (e.g. Z-score or beta estimates, regression estimates from linear associations?)

Figure 1: axis are very hard to read, maybe better to move this to Supplementary or show a subset.

Line 201: unless this is the only putative causal SNP and mechanism, this sentence seems out of context since PATZ1 is not even shown here. This section could use a rearrangement.

Line 236: "Group A" and "Group B" are mentioned but not explained here, remove or explain.

Line 241: Do the GTEx effect sizes include non-European individuals? If so, while the LD uses CEU individuals, it would be good to add a word of caution for this in the Discussion.

Line 269: The authors make a connection between their results on the direction of effect in the FADS1 and FADS2 genes, and the fact that it is a duplicated gene. The importance of this observation in light of copy number variants is reiterated in the Discussion (line 390). I am having trouble understanding how the results point to the duplication/copy numbers from effect direction, can the authors clarify this?

Line 285: is the frequency of Hi-C fragment contact counts indicating the number of reads in support of an interaction, or the number of distinct interactions? In either case, can the other case be shown? E.g. correlation between strength of eQTL associations and [number of reads supporting the interactions] or [number of independent loops] ?

Line 317-320: The high proportion of genes possibly with mappability uncertainty in the linkage group could possibly be checked by looking at lists of known pseudo genes for example available from Ensembl.

Line 341: slight correction to this sentence? e.g. have been missed by proximity to GWAS associations.

Reviewer #3 (Remarks to the Author):

The authors have carefully addressed all my initial concerns. I have no further comments to add.

Dear Editor,

Thank you and the reviewers for your constructive comments that have enabled us to further improve our manuscript. We have provided more details in the methods section, performed an additional analysis, and made changes to the relevant sections of the manuscript in response to your comments. Please find below a detailed list of the improvements done.

We strongly believe that this second revision has significantly improved our study and manuscript. We look forward to hearing from you.

Sincerely

Justin M. O'Sullivan

Reviewer #2 (Remarks to the Author):

1) Which pipelines were used in the CoDeS3D pipeline to get the raw interaction HiC data (e.g. HiC-Pro)?

⇒ We thank the reviewer for raising this important point. For this study, CoDeS3D used pre-processed Hi-C mappings from Rao et al (2014). We have reproduced this de-duplicated Hi-C pair mappings from the raw Hi-C reads of the Rao et al (2014) cells and a number of other Hi-C experiments using pipelines like Juicer. This is outlined in the *Identification of spatial SNP-gene pairs* section of **Methods**:

“Rao et al²¹ have previously prepared high resolution Hi-C chromatin interaction libraries of the seven cell lines (*i.e.* GM12878, HMEC, HUVEC, IMR90, K562, KBM7 and NHEK) used in this study. In summary, they used an in-house pipeline that included BWA alignment of paired end reads against the hg19 build of the human genome. They then filtered out “ambiguous alignments”, difficult to align read pairs, and duplicates. We downloaded the cleaned mapping locations of Hi-C read pairs (*i.e.* HIC*_merged_nodups.txt files) from Gene Expression Omnibus (GEO accession number = GSE63525, *S6 Table) using the “download_default_data” module in CoDeS3D. The required data format is such that the rows in the interaction file describe the alignment of both read pairs (*i.e.* 1 and 2) with the following columns: read name, strand1, chromosome1, position1, fragment1, strand2, chromosome2, position2, fragment2. CoDeS3D works with any Hi-C pipeline that can generate interaction files in the required data format (*e.g.* HOMER, Juicer).

The hg19 human genome sequence was digested with the same restriction enzyme employed in preparing the Hi-C libraries (*i.e.* Mbol) using the Restriction package of Biopython in the “digest_genome” module of CoDeS3D. The genome digestion produced a table of DNA fragments, which is queried by the “process_inputs” and “find_interactions” modules of CoDeS3D to identify the fragments harbouring SNPs, and their paired fragments. Next, “CoDeS3D.find_genes” was used with the “intersect” command in pybedtools to identify paired fragments that overlap with a composite gene

region within the reference genome (Fig 1a). There was no binning or padding around restriction fragments to obtain gene overlap. Genes within, or overlapping, restriction fragments, which are in contact with fragments containing the daSNPs were identified as spatial pairs to the SNP-containing fragments.”

2) How were the HiC interaction data intersected with "genes" (please define "genes", is it from promoter regions with padded number of bases? or using the GTEx definition?)

⇒ We used the GTEx definition of a gene, which is a collapse of GENCODE isoforms into single genes. The text in the *Data and reference files* section of **Methods** now reflects this:

“We used the GENCODE transcript model (See Data Availability) as the reference for gene annotations. The GENCODE transcript model is also used in GTEx. All isoforms of a gene were collapsed into a single composite gene region.”

3) In Methods specify that you are using the GTEx results from both the cis and trans eQTLs

⇒ We have made this change. The *Identification of spatial eQTL-eGene pairs* section in **Methods** now reads:

“The resulting SNP-gene pairs were then used to query the GTEx database (www.gtexportal.org, multi-tissue eQTLs analysis v4) to identify eQTL-eGene pairs *i.e.* SNPs that are associated with a change in the expression genes in at least one tissue. The “Test Your Own” API on the GTEx portal allows for the analysis of both cis and trans eQTL associations. Spatial SNP-gene pairs that have no eQTL association are excluded from the subsequent steps, which include multiple testing. The false discovery rate of the eQTL associations p-values were adjusted using the Benjamini-Hochberg procedure, and associations with adjusted p-values < 0.05 were deemed spatial eQTL-eGene pairs.”

4) The authors say that they used unfiltered raw interaction files, using no binning or padding. But isn't the raw interaction data from most pipelines defined at particular resolutions?

⇒ We used Hi-C libraries from Rao because they have the highest fidelity in terms of sequencing depth and choice of restriction enzyme, which is why they were able to achieve 1kb resolution contact matrices. As discussed in 1) above, we used Hi-C mappings after the fragment-level filtering. We did not construct contact matrices, which is the step in Hi-C where binning is used. For our use, interactions at the fragment level offer the most utility in this discovery-based approach to answering the question of SNP-gene interactions.

5) Also, it seems as the authors used all raw interactions from Hi-C with no cutoff for FDR or number of reads, so some of these loops could be supported by only one read, is that correct? I understand that the loops are then filtered using the eQTL dataset. However, loops supported by only one read have a very high chance of being false positives. It would be good to check whether the loops with lower number of reads lack support from eQTLs and are therefore filtered out in any case.

⇒ It is true that we did not use any threshold for the number of pair reads. This is because the fragment-level mappings contain only unique pair reads after filtering out abnormal alignments, duplicates, and low-quality alignments. A threshold would be most appropriate had we binned the datasets. In that case, the threshold would be based on the number of interactions between bins. In this situation, however, we used number of interactions between a SNP fragment and any fragment(s) within a gene. We have clarified this in Fig 1a.

“A spatial SNP-gene pair is defined as interaction(s), in ≥ 1 cell line(s), between the fragment harbouring the SNP and any fragment(s) in the region spanning the gene. In the cartoon example in (a) there are two unique fragment interactions between the SNP and gene fragments. One interaction is captured in only one replicate, R1, of cell line, CL1. The second interaction is captured in two replicates, R1 and R2, of cell line CL1, and one replicate, R1, of cell line CL2. Thus, the SNP-gene pair has a total of two fragment interactions, and four supporting interactions.”

⇒ We have also confirmed that SNP-gene pairs with low number of supporting interactions are filtered out at the eQTL step. This is shown in Fig 1b, and we have included a sentence in the first paragraph of *GWAS SNPs mark spatial regulatory regions* section in **Results** to that effect:

“Spatial SNP-gene pairs with evidence of interaction in >1 cell lines or >1 replicates in a single cell line are significantly more enriched (two-proportions Z-test, p-value $< 2.2 \times 10^{-16}$) for eQTL associations than pairs with only one interaction in one replicate of a cell line (Fig 1b).”

6) Abstract/intro: Some inconsistency in wording could be solved by defining in the Introduction a spatial eQTL-eGene pair as a daSNP-gene pair which is supported by both interaction and eQTL data, and using the same working throughout.

⇒ Thank you for the observation. We have corrected this throughout the manuscript.

7) Figure 1: are the phenotypes with more number of reported daSNPs better represented? If this is the case, please specify in Discussion for caveats.

⇒ Indeed, the phenotypes with greater number of daSNPs generally have more eGenes (correlation coefficient $r = 0.89$). We have included the following in the text in paragraph 3 in **Discussion**:

“...The correlation ($r = 0.89$) between number of daSNPs and spatial eGenes suggests that phenotypes with greater number of associated SNPs are better represented in pleiotropy.”

8) Line 96: out of the 1,528 trans-interactions, how many are on different chromosomes? Please specify the number for these here.

⇒ 865 eGenes are affected by an eQTL from a different chromosome. A sentence in paragraph 3 in the *GWAS SNPs mark spatial regulatory regions* section of Results now reads thus:

“... Of the 7,917 affected eGenes, 70.0% (5,545) were associated with only *cis*-spatial interactions (*i.e.* both partners are from the same chromosome and separated by <1,000,000 bp, as defined elsewhere^{24,25}), 19.3% (1,528) by only *trans*-interactions

(*i.e.* 663 are affected by an eQTL SNP on the same chromosome but separated by $\geq 1,000,000$ bp, and 865 are affected by an eQTL SNP from different chromosomes^{24,25}), and 10.7% (844) by both *cis*- and *trans*-interactions (Fig 1c, a in S2 Fig, S1 Table).”

9) Lines 101-102: can remove since it is already in Supp Fig 2C.

⇒ Removed.

10) Line 114 and Line 284: how were the strength of eQTL associations measured (e.g. Z-score or beta estimates, regression estimates from linear associations?)

⇒ The eQTL effect sizes are normalized effect sizes (*i.e.* beta estimates), while the strength of associations is the eQTL p-values. We have clarified these in the text:

“... 866 (19.6%) of the ‘GTEx only’ associations, with eQTL normalised effect sizes (NES *i.e.* the slope of linear regression) of -1.11 to 1.15, were lost in the CoDeS3D analysis due to lack of evidence for a spatial connection between the genomic fragments containing the SNP and gene.” (Paragraph 4 in the *GWAS SNPs mark spatial regulatory regions* section of **Results**)

“There was no correlation between the total strength (*i.e.* p values) of eQTL associations and the total number of spatial interactions (a and b in S10).” (Paragraph 3 in the eQTLs have gene and tissue specific effect patterns of **Results**)

11) Figure 1: axis are very hard to read, maybe better to move this to Supplementary or show a subset.

⇒ We have moved Figure 1 to the supplementary materials as suggested.

12) Line 201: unless this is the only putative causal SNP and mechanism, this sentence seems out of context since PATZ1 is not even shown here. This section could use a rearrangement.

⇒ Thank you for pointing this out. We have modified the text to read:

We mapped the spatial eQTL-eGene interactions within the *FADS1-3* locus (Fig 3a) in order to investigate the effect of genetic variation on the regulatory network for the multimorbid phenotypes associated with the cluster. The transcription levels of the *FADS1*, *FADS2*, *TMEM258*, and *DAGLA* genes, that are central to this cluster, are associated with eQTLs that are located within these genes and across the region (Fig 3a). Nine of the putative regulatory regions are located within introns of genes (*i.e.* *MYRF*, *TMEM258*, *FEN1*, *FADS1*, *FADS2* and *FADS3*) in this locus. Putative regulatory effects linking eQTLs in *FADS1-3* with *DAGLA*, or eQTLs in *MYRF* with *FADS1-3* cross a topologically associating domain (TAD) boundary located in the vicinity of *FEN1*, whose transcription is not associated with any of the eQTLs (S7 Fig). Spatial eQTLs associated with some phenotypes (*e.g.* LDL cholesterol, muscle measurement and comprehensive strength index) are few and localised while others (*e.g.* *cis-trans*-18:2 fatty acid, phospholipid) are dispersed across the locus. However, despite this, almost all of the phenotype-associated SNPs in this cluster are correlated with a change in the transcript level of more than one gene (Fig 3b). We also observed this one-to-many SNP-eGene eQTL association in the pulmonary cluster, including inter-TAD connections from the region marked by rs8042374 (c in S9 Fig).

Collectively, these results are consistent with previous reports of the formation of complex networks of multiple long-range interactions by regulatory elements and gene promoters⁴⁴.

13) Line 236: "Group A" and "Group B" are mentioned but not explained here, remove or explain.

⇒ Thank you for identifying this error. We have corrected it.

“...eQTL SNPs are coloured according to the putative regulatory regions in (a).”

14) Line 241: Do the GTEx effect sizes include non-European individuals? If so, while the LD uses CEU individuals, it would be good to add a word of caution for this in the Discussion.

⇒ The referee is correct, GTEx includes non-European individuals. The text has been modified:

“... The effects of these composite regulatory elements are reflected in the LD architecture of the regions. The LD data was from the 1000 Genomes CEU population (whereas the GTEx population is 85.2% Caucasian), however the results indicate that the inheritance of these regions may be linked to physical association between regions that are separated in the linear sequence.”

15) Line 269: The authors make a connection between their results on the direction of effect in the FADS1 and FADS2 genes, and the fact that it is a duplicated gene. The importance of this observation in light of copy number variants is reiterated in the Discussion (line 390). I am having trouble understanding how the results point to the duplication/copy numbers from effect direction, can the authors clarify this?

⇒ We thank the reviewer for highlighting an area where we over-interpreted our data. We have removed references to the copy number from the manuscript.

16) Line 285: is the frequency of Hi-C fragment contact counts indicating the number of reads in support of an interaction, or the number of distinct interactions? In either case, can the other case be shown? E.g. correlation between strength of eQTL associations and [number of reads supporting the interactions] or [number of independent loops] ?

⇒ Fig S10 (a and b) has been modified and now shows correlations for both the distinct and supporting interactions (as defined in Fig 1a and addressed in 5 above).

17) Line 317-320: The high proportion of genes possibly with mappability uncertainty in the linkage group could possibly be checked by looking at lists of known pseudo genes for example available from Ensembl.

⇒ We have checked this. We did not find matches for these genes in Ensembl either.

18) Line 341: slight correction to this sentence? e.g. have been missed by proximity to GWAS associations.

⇒ We have corrected this error.

Reviewer #2 (Remarks to the Author):

The authors have carefully addressed and satisfactorily responded to all my questions.